# Local Augmentation for Graph Neural Networks

## Abstract

Data augmentation has been widely used in image data and linguistic data but remains under-explored for Graph Neural Networks (GNNs). Existing methods focus on augmenting the graph data from a global perspective and largely fall into two genres: structural manipulation and adversarial training with feature noise injection. However, recent graph data augmentation methods ignore the importance of local information for the GNNs' message passing mechanism. In this work, we introduce the local augmentation, which enhances the locality of node representations by their subgraph structures. Specifically, we model the data augmentation as a feature generation process. Given a node's features, our local augmentation approach learns the conditional distribution of its neighbors' features and generates more neighbors' features to boost the performance of downstream tasks. Based on the local augmentation, we further design a novel framework: LA-GNN, which can apply to any GNN models in a plug-and-play manner. Extensive experiments and analyses show that local augmentation consistently yields performance improvement for various GNN architectures across a diverse set of benchmarks.

## 1 Introduction

Graph Neural Networks (GNNs) and their variants (Abu-El-Haija et al., 2019; Kipf & Welling, 2017; Veličković et al., 2018) have achieved state-of-the-art performance for many tasks on graphs such as recommendation system (Ying et al., 2018) and traffic prediction (Guo et al., 2019). However, most of the GNN models, such as GCN (Kipf & Welling, 2017) and GAT (Veličković et al., 2018), learn the node representations by aggregating information over only the 2-hop neighborhood. Such shallow architectures limit their ability to extract information from higher-layer neighborhoods (Wang & Derr, 2021). But deep GNNs are prone to over-smoothing (Li et al., 2018), which suggests the node representations tend to converge to a certain vector and thus become indistinguishable. One solution to address this problem is to preserve the locality of node representations when increasing the number of layers. For example, JKNet (Xu et al., 2018) densely connects (Huang et al., 2017) each hidden layer to the final layer. GCNII (Chen et al., 2020) employs an initial residual to construct a skip connection from the input layer. Besides, Zeng et al. (2021) pointed out that the key for GNN is to smooth the local neighborhood into informative representation, no matter how deep it is. And they decouple the depth and scope of GNNs to help capture local graph structure. Prior works have emphasized the importance of local information, but one property of the graph is that the number of nodes in the local neighborhood is far fewer than higher-order neighbors. And this property limits the expressive power of GNNs due to the limited neighbors in the local structure. A very intuitive idea is to use data augmentation to increase the number of nodes in the local substructure.

However, existing graph data augmentation methods ignore the importance of local information and only perturb at the topology-level and feature-level from a global perspective, which can be divided into two categories: topology-level augmentation (Rong et al., 2020; Wang et al., 2020b; Zhao et al., 2021) and feature-level augmentation (Deng et al., 2019; Feng et al., 2019; Kong et al., 2020). Topology-level augmentation perturbs the adjacency matrix, yielding different graph structures. On the other hand, existing feature-level augmentation mainly exploits perturbation of node attributes guided by adversarial training (Deng et al., 2019; Feng et al., 2019; Kong et al., 2020). These augmentation techniques have two drawbacks. 1) Some of they employ full-batch training for augmentation, which is computationally expensive, and introduce some additional side effects such

as over-smoothing. 2) The type of feature-level augmentation is coarse-grained, which focuses on global augmentation and overlooks the local information of the neighborhood. Moreover, to our best knowledge, none of the existing approaches combines both the feature representations and the graph topology, especially the local subgraph structures, for graph-level data augmentation.

In this work, we propose a framework: **L**ocal **A**ugmentation for **G**raph **N**eural **N**etworks (LA-GNNs), to further enhance the locality of node representations based on both the topology-level and feature-level information in the substructure. The term "local augmentation" refers to the generation of neighborhood features via a generative model **conditioned on local structures and node features**. Specifically, our proposed framework learns the conditional distribution of the connected neighbors' representations given the representation of the central node, bearing some similarities with the Skip-gram (Mikolov et al., 2013) and Deepwalk Perozzi et al. (2014), with the difference that our method does not base on word or graph embedding.

The motivation behind this work concludes three-fold. 1) Existing feature-level augmentation works primarily pay attention to global augmentation without considering the informative neighborhood. 2) The distributions of the representations of the neighbors are closely connected to the central node, making ample room for feature augmentation. 3) Preserving the locality of node representations is key to avoiding over-smoothing (Xu et al., 2018; Klicpera et al., 2019; Chen et al., 2020). And there are several benefits in applying local augmentation for the GNN training. First, local augmentation is essentially a data augmentation technique that can improve the generalization of the GNN models and prevent over-fitting. Second, we can recover some missing contextual information of the local neighborhood in an attributed graph via the generative model (Jia & Benson, 2020). Third, our proposed framework is flexible and can be applied to various popular backbone networks such as GCN (Kipf & Welling, 2017), GAT (Veličković et al., 2018), GCNII (Chen et al., 2020), and GRAND (Feng et al., 2020) to enhance their performance. Extensive experimental results demonstrate that our proposed framework could improve the performance of GNN variants on 7 benchmark datasets.

## 2 BACKGROUND

**Notations.** Let $G = (V, E)$ represent the graph, where $V$ is the set of vertices $\{v_1, \cdots, v_N\}$ with $|V| = N$ and $E$ is the set of edges. The adjacency matrix is defined as $\mathbf{A} \in \{0, 1\}^{N \times N}$, and $\mathbf{A}_{ij} = 1$ if and only if $(v_i, v_j) \in E$. Let $\mathcal{N}_i = \{v_j | \mathbf{A}_{ij} = 1\}$ denotes the neighborhood of node $v_i$ and $\mathbf{D}$ denote the diagonal degree matrix, where $\mathbf{D}_{ii} = \sum_{j=1}^n \mathbf{A}_{ij}$. The feature matrix is denoted as $\mathbf{X} \in \mathbb{R}^{N \times F}$ where each node $v$ is associated with a $F$-dimensional feature vector $\mathbf{X}_v$. $\mathbf{Y} \in \{0, 1\}^{N \times C}$ denote the one-hot label matrix, where $\mathbf{Y}_i \in \{0, 1\}^C$ is a one-hot vector and $\sum_{j=1}^C \mathbf{Y}_{ij} = 1$ for any $v_i \in V$.

**GNN.** Graph Neural Network (GNN) is a type of neural network that directly operates on the graph structure, such as GCN and GAT (Kipf & Welling, 2017; Veličković et al., 2018), that capture the dependence of graphs via message passing between the nodes of a graph as

$$\mathbf{H}^{(\ell)} = f(\mathbf{A}, \mathbf{H}^{(\ell-1)}), \tag{1}$$

where $f$ denotes the specific GNN layer for different models, $\mathbf{H}^{(\ell)}$ are the hidden vectors of the $\ell$-th layer and $\mathbf{H}^{(0)} = \mathbf{X}$. For example, $f(\mathbf{A}, \mathbf{H}) = \sigma(\hat{\mathbf{A}}\mathbf{H}\mathbf{W})$ for GCN, where $\hat{\mathbf{A}} = \tilde{\mathbf{D}}^{-\frac{1}{2}}\tilde{\mathbf{A}}\tilde{\mathbf{D}}^{-\frac{1}{2}}$, $\tilde{\mathbf{D}}$ is the degree matrix of $\tilde{\mathbf{A}}$, i.e., $\tilde{\mathbf{D}}_{ii} = \sum_j \tilde{\mathbf{A}}_{ij}$, and $\tilde{\mathbf{A}} = \mathbf{A} + \mathbf{I}$.

**Topology-level Augmentation.** Topology-level augmentation usually perturbs $\mathbf{A}$ to generate different graph structures, which can be formulated as $\mathbf{A}' = \mathcal{F}(\mathbf{A}, \mathbf{X})$, where $\mathcal{F}(\cdot)$ is a structure perturbation function. For example, DropEdge (Rong et al., 2020) considers $\mathcal{F}(\mathbf{A}, \mathbf{X}) = \mathbf{A} - \mathbf{A_s}$ which is independent of $\mathbf{X}$, where $\mathbf{A_s}$ is a sparse matrix consists of a subset of the original edges $E$. GAUG-O (Zhao et al., 2021) leverages their proposed neural edge predictors to produce a different structure $\mathbf{A}'$ where $\mathbf{A}'_{ij} = \left\lfloor \frac{1}{1+e^{-(\log \mathbf{P}_{ij}+G)/\tau}} + \frac{1}{2} \right\rfloor$, $\mathbf{P}_{ij} = \alpha \mathbf{M}_{ij} + (1-\alpha)\mathbf{A}_{ij}$, $\mathbf{M} = \sigma\left(\mathbf{Z}\mathbf{Z}^T\right)$, $\mathbf{Z} = f\left(\mathbf{A}, f(\mathbf{A}, \mathbf{X})\right)$, $\tau$ is the temperature of Gumbel-Softmax distribution, $G \sim \text{Gumbel}(0, 1)$ is a Gumbel random variate, and $\alpha$ is a hyperparameter mediating the influence of edge predictor on the original graph.

**Feature-level Augmentation.** Besides, feature-level augmentation function can be defines as $\mathbf{X}' = \mathcal{H}(\mathbf{A}, \mathbf{X})$, where $\mathcal{H}(\cdot)$ is a feature perturbation function. FLAG (Kong et al., 2020) defines the perturbation function as $\mathcal{H}(\mathbf{A}, \mathbf{X}) = \mathbf{X} + \boldsymbol{\delta}$ where

Table 1: Comparison of existing graph data augmentation.

| Graph Data Augmentation | | | |
|---|---|---|---|
| Method | Considered Part | Type | Perturbed Part |
| DropEdge | $\mathbf{A}$ | Sampling | $\mathbf{A}$ |
| GAUG-O | $\mathbf{A}\&\mathbf{X}$ | Reconstruction | $\mathbf{A}$ |
| FLAG | $\mathbf{X}$ | Noise Injection | $\mathbf{X}$ |
| G-GCN | $\mathbf{A}\&\mathbf{X}$ | Reconstruction | $\mathbf{X}$ |
| Local Augmentation | $\mathbf{A}\&\mathbf{X}$ | Generation | $\mathbf{X}$ |

perturbation $\boldsymbol{\delta}$ is updated iteratively during the adversarial training phase. G-GCN (plain) (Zhu et al., 2020) obtains the global attribute feature matrix $\mathbf{X}^{(a)} \in \mathbb{R}^{N \times d_a}$ through minimizing the objective $\prod_{v \in V} \prod_{a \in CA(v)} \frac{\exp\left(\mathbf{X}_v^{(a)} \cdot \mathbf{V}_a\right)}{\sum_{k \in U} \exp\left(\mathbf{X}_v^{(a)} \cdot \mathbf{V}_k\right)}$ where $U$ is the set of all attributes, $CA(v)$ is the sampled context attributes of $v$, and $\mathbf{V} \in \mathbb{R}^{d_a \times F}$ denotes the parameters. Obviously, the perturbation function of G-GCN has no close-form solution. In this work, we propose a novel feature-level augmentation method, named local augmentation. And the comparison of the details of various graph data augmentation techniques can be found in Table 1.

## 3 LOCAL AUGMENTATION

In this section, we describe details of the proposed method. The local augmentation framework consists of three modules: learning the conditional distribution via a generative model, the active learning trick, and the downstream GNN models, as illustrated in Figure 1. Note that the proposed algorithm enhances the locality of node representations through augmenting 1-hop neighbors in a generative way. Specifically, we exploit a generative model to learn the conditional distribution of the connected neighbors' representations given the representation of a node. We describe the details of learning the conditional distribution and the motivation for why local augmentation is able to improve the performance in a probabilistic view in Sec. 3.1, detail the architecture of downstream GNN models in Sec. 3.2. We finally elaborate the training procedure of both the generative model and the downstream GNN models with the active learning trick in Sec. 3.3.

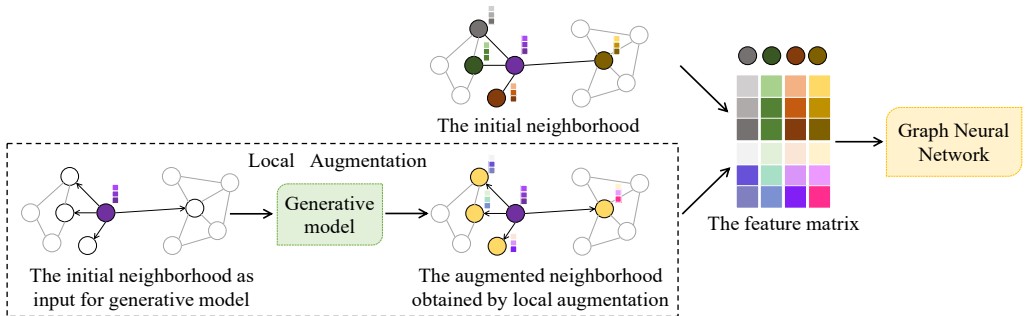

Figure 1: A schematic depiction of our local augmentation. The purple and yellow circles on the graph correspond to the central node and its augmented neighbors respectively. After augmenting the neighborhood, we exploit the initial and the generated feature matrix as input for downstream GNNs.

### 3.1 LEARNING THE CONDITIONAL DISTRIBUTION

We start by reviewing the semi-supervised learning of GNNs in a probabilistic view. Most existing GNN models (Kipf & Welling, 2017; Veličković et al., 2018) are viewed as a classification function to predict the class labels of the graph nodes. In this work, we use a GNN classification estimator $P_\theta(\mathbf{Y}|\mathbf{A}, \mathbf{X})$ ($\theta$ is the parameter) to model the conditional distribution of label $\mathbf{Y}$ with respect to the graph structure $\mathbf{A}$ and feature matrix $\mathbf{X}$. Given training samples $\{\mathbf{A}, \mathbf{X}, \mathbf{Y}\}$, the parameter $\theta$ can be estimated using Maximum Likelihood Estimation (MLE), by optimizing the following likelihood function:

$$\max \prod_{k \in \mathbf{K}} P_\theta\left(\mathbf{Y}_k | \mathbf{A}, \mathbf{X}\right), \tag{2}$$

where $\mathbf{K}$ is the set of node indices of the training dataset whose labels are visible during the semi-supervised training. To further boost the performance of GNN, we introduce a new model $P_\theta(\mathbf{Y}, \overline{\mathbf{X}}|\mathbf{A}, \mathbf{X})$, where $\overline{\mathbf{X}}$ is generated features by feature-level augmentation. For this model, the MLE method needs to optimize a marginalized probability $P_\theta$ over the generated feature matrix $\overline{\mathbf{X}}$:

$$\max \prod_{k \in \mathbf{K}} \int_{\overline{\mathbf{X}}} P_\theta\left(\mathbf{Y}_k, \overline{\mathbf{X}}|\mathbf{A}, \mathbf{X}\right). \tag{3}$$

For Bayesian tractability, we decompose $P_\theta$ in Eq.(3) as a product of two posterior probabilities:

$$P_{\theta,\phi}(\mathbf{Y}_k, \overline{\mathbf{X}}|\mathbf{A}, \mathbf{X}) := P_\theta(\mathbf{Y}_k|\mathbf{A}, \mathbf{X}, \overline{\mathbf{X}})Q_\phi(\overline{\mathbf{X}}|\mathbf{A}, \mathbf{X}), \tag{4}$$

where $P_\theta(\mathbf{Y}_k|\mathbf{A}, \mathbf{X}, \overline{\mathbf{X}})$ and $Q_\phi(\overline{\mathbf{X}}|\mathbf{A}, \mathbf{X})$ denote the probabilistic distributions approximated by the downstream GNN and the (feature-level augmentation) generator respectively, parameterized by $\theta$ and $\phi$. **There are two benefits in the decomposition above.** First, it allows us to decouple the training of the downstream predictor $P_\theta$ and the generator $Q_\phi$, enabling the generator to easily generalize to other downstream tasks. Moreover, inspired by the successes of data augmentation via deep-learning-based generative modeling (Antoniou et al., 2017), the representation power of Eq.(4) is superior than that of a single predictor $P_\theta(\mathbf{Y}_k|\mathbf{A}, \mathbf{X})$ without data augmentation.

Consequently, once a generator $Q_\phi$ is trained very well, our training procedure can optimize $P_\theta(\mathbf{Y}_k|\mathbf{A}, \mathbf{X}, \overline{\mathbf{X}})$ with samples $\overline{\mathbf{X}}$ drawn from the fixed conditional distribution $Q_\phi$. Now, we show how to train the generator as follows.

**Generator** To learn a feature augmentation generator, a naive solution is to learn one single distribution for all the neighbors using the MLE method, i.e., solving the following optimization problem

$$\max_\psi \sum_{j \in \mathcal{N}_i} \log p_\psi\left(\mathbf{X}_j|\mathbf{X}_i\right) = \max_\psi \log \prod_{j \in \mathcal{N}_i} p_\psi\left(\mathbf{X}_j|\mathbf{X}_i\right), \tag{5}$$

where $\{\mathbf{X}_{j|j \in \mathcal{N}_i}, \mathbf{X}_i\}$. Then $p_\psi$ can be used to augment features for all the neighbors. However, this method ignores the differences between all the neighbors, which may induce severe noise.

To overcome the limitation, we assume that each neighbor satisfies a different conditional distribution. Specifically, there exists a conditional distribution $p(\cdot|\mathbf{X}_i, \mathbf{z}_j)$ with latent random variable $\mathbf{z}_j$, such that we have $\mathbf{X}_j \sim p(\mathbf{X}|\mathbf{X}_i, \mathbf{z}_j)$ for $\mathbf{X}_{j|j \in \mathcal{N}_i}$. Once we obtain $p(\cdot|\mathbf{X}_i, \mathbf{z}_j)$ in some way, we can generate augmented features $\overline{\mathbf{X}}$, and then we can train $P_\theta(\mathbf{Y}_k|\mathbf{A}, \mathbf{X}, \overline{\mathbf{X}})$ instead of $P_\theta(\mathbf{Y}_k|\mathbf{A}, \mathbf{X})$ to improve the final performance of $P_\theta$. Below, we will present how to find $p(\cdot|\mathbf{X}_i, \mathbf{z}_j)$, which will produce the generator $Q_\phi$.

To achieve our purpose, a suitable method is the conditional variational auto-encoder (CVAE) (Kingma & Welling, 2013; Sohn et al., 2015), which can help learn the distribution of the latent variable $\mathbf{z}_j$, and the conditional distribution $p(\cdot|\mathbf{X}_i, \mathbf{z}_j)$. So, a CVAE model $Q_\phi\left(\overline{\mathbf{X}}|\mathbf{A}, \mathbf{X}\right)$ is adopted as our generator, where $\phi = \{\varphi, \psi\}$, $\varphi$ denotes the variational parameters and $\psi$ represents the generative parameters. To derive the optimization problem for CVAE, $\log p_\psi\left(\mathbf{X}_j|\mathbf{X}_i\right)$ can be written with latent variables $\mathbf{z}$ as follows, following previous work (Pandey & Dukkipati, 2017; Sohn et al., 2015):

$$\log p_\psi\left(\mathbf{X}_j|\mathbf{X}_i\right) = \int q_\varphi(\mathbf{z}|\mathbf{X}_j, \mathbf{X}_i) \log \frac{p_\psi(\mathbf{X}_j, \mathbf{z}|\mathbf{X}_i)}{q_\varphi(\mathbf{z}|\mathbf{X}_j, \mathbf{X}_i)} d\mathbf{z} + KL(q_\varphi(\mathbf{z}|\mathbf{X}_j, \mathbf{X}_i)\|p_\psi(\mathbf{z}|\mathbf{X}_j, \mathbf{X}_i))$$

$$\geq \int q_\varphi(\mathbf{z}|\mathbf{X}_j, \mathbf{X}_i) \log \frac{p_\psi(\mathbf{X}_j, \mathbf{z}|\mathbf{X}_i)}{q_\varphi(\mathbf{z}|\mathbf{X}_j, \mathbf{X}_i)} d\mathbf{z},$$

and the evidence lower bound (ELBO) can be written as:

$$\mathcal{L}(\mathbf{X}_j, \mathbf{X}_i; \psi, \varphi) = -KL(q_\varphi(\mathbf{z}|\mathbf{X}_j, \mathbf{X}_i)\|p_\psi(\mathbf{z}|\mathbf{X}_i)) + \int q_\varphi(\mathbf{z}|\mathbf{X}_j, \mathbf{X}_i) \log p_\psi(\mathbf{X}_j|\mathbf{X}_i, \mathbf{z}) d\mathbf{z}, \tag{6}$$

where the encoder $q_\varphi(\mathbf{z}|\mathbf{X}_j, \mathbf{X}_i) = \mathcal{N}(f(\mathbf{X}_j, \mathbf{X}_i), g(\mathbf{X}_j, \mathbf{X}_i))$ and decoder $p_\psi(\mathbf{X}_j|\mathbf{X}_i, \mathbf{z}) = \mathcal{N}(h(\mathbf{X}_i, \mathbf{z}), cI)$. The encoder is a two-layer MLP. $f$ and $g$ share the first layer, and their second layers employ different parameters. The decoder $h$ is two-layer MLP. For simplicity and tractability, the implemented generator $Q\left(\overline{\mathbf{X}}|\mathbf{A}, \mathbf{X}\right)$ uses the same parameters across all nodes $v_i \in V$.

**Optimization of the MLE**   Now, we present how to optimize the MLE Eq.(4) using the feature matrix produced from the generator. Once the augmented feature matrix can be sampled from the generator, we can optimize the parameters of Eq.(4) in the following way. Firstly, the parameter $\phi = \{\psi, \varphi\}$ can be optimized by maximizing the ELBO of the generator (6), i.e., we train the generator. Secondly, the parameter $\theta$ is optimized by maximizing the MLE Eq.(4) with $\phi$ fixed, which is the conditional distribution of $\mathbf{Y}_k$ given $\mathbf{A}$, $\mathbf{X}$, and $\overline{\mathbf{X}}$, i.e., we train the downstream GNN model.

In this paper, the MLE is formulated by a downstream GNN model as follows:

$$P_\theta\left(\mathbf{Y}_k \mid \mathbf{A}, \mathbf{X}, \overline{\mathbf{X}}\right) \propto -\overline{\mathcal{L}}(\theta|\mathbf{A}, \mathbf{X}, \overline{\mathbf{X}}, \phi), \tag{7}$$

where $\overline{\mathcal{L}}(\theta|\mathbf{A}, \mathbf{X}, \overline{\mathbf{X}}, \phi) = -\sum_{k \in \mathbf{T}} \sum_{f=1}^{C} \mathbf{Y}_{kf} \ln\left(\text{softmax}\left(\text{GNN}(\mathbf{A}, \mathbf{X}, \overline{\mathbf{X}})\right)_{kf}\right).$

### 3.2   THE ARCHITECTURE OF LA-GNN

We discuss the details of downstream GNN models. And we use GCN, GAT, GCNII, and GRAND as the backbones and test them on semi-supervised node classification tasks. We name the modified GNN architecture as LA-GNN, where LA means local augmentation.

**LA-GCN**   A 2-layer LA-GCN is defined as follows:

$$\mathbf{H}^{(2)} = \sigma\left(\hat{\mathbf{A}}\left(\sigma\left(\hat{\mathbf{A}}\mathbf{X}\mathbf{W}_1^{(1)}\right)\middle\|\sigma\left(\hat{\mathbf{A}}\overline{\mathbf{X}}_1\mathbf{W}_2^{(1)}\right)\middle\|\cdots\middle\|\sigma\left(\hat{\mathbf{A}}\overline{\mathbf{X}}_n\mathbf{W}_{n+1}^{(1)}\right)\right)\mathbf{W}^{(2)}\right), \tag{8}$$

where $\overline{\mathbf{X}}_i$ $(i = 1, 2, \cdots, n)$ is the augmented feature matrix produced by the generator, $\|$ denotes an operator of column-wise concatenation, $\mathbf{W}_i^{(1)}$ $(i = 1, 2, \cdots, n)$ denotes the parameters of the first LA-GCN layer, and $\mathbf{W}^{(2)}$ denotes the parameters of the second LA-GCN layer.

**LA-GCNII**   Since GCNII (Chen et al., 2020) applies a fully-connected neural network on $\mathbf{X}$ to obtain a lower-dimensional initial representation $\mathbf{H}^{(0)}$ before the forward propagation, we apply a fully-connected neural network on $\mathbf{X}$ and $\overline{\mathbf{X}}$ to obtain $\mathbf{H}^{(0)}$ for LA-GCNII as follows:

$$\mathbf{H}^{(0)} = \sigma\left(\mathbf{X}\mathbf{W}_1^{(0)}\right)\middle\|\sigma\left(\overline{\mathbf{X}}_1\mathbf{W}_2^{(0)}\right)\middle\|\cdots\middle\|\sigma\left(\overline{\mathbf{X}}_n\mathbf{W}_{n+1}^{(0)}\right). \tag{9}$$

$\mathbf{H}^{(0)}$ is fed into the next forward propagation layer. Besides, we do not modify the architecture of GAT and GRAND, and just add our generated feature matrix to the input.

### 3.3   ACTIVE LEARNING

In this section, we introduce a trick for the overall training framework. After the training of the generator finishes, it contains an issue of using $Q_\phi(\overline{\mathbf{X}}|\mathbf{A}, \mathbf{X})$ of Eq.(4) for inference because $Q$ may generate some samples from the side part of the distribution. This critical question makes the inferences inefficient. Inspired by Nielsen & Okoniewski (2019), we introduce active learning to capture the suitable generated feature matrix and the corresponding generator, which improves the inference efficiency and helps the optimization of the MLE. During active learning, the probability of each feature is proportional to its uncertainty evaluated by an acquisition function. We adopt the Bayesian Active Learning by Disagreement (BALD) acquisition function (Houlsby et al., 2011) to sample the most important inferences with the approximation from the Monte Carlo (MC) dropout samples as

$$U(\overline{\mathbf{X}}) \approx H\left[\frac{1}{N}\sum_{n=1}^{N} P\left(\mathbf{Y}_k|\overline{\mathbf{X}}, \boldsymbol{\omega}_n\right)\right] - \frac{1}{N}\sum_{n=1}^{N} H\left[P\left(\mathbf{Y}_k|\overline{\mathbf{X}}, \boldsymbol{\omega}_n\right)\right], \tag{10}$$

where $N$ is the number of MC samples and $\boldsymbol{\omega}_n$ are the parameters of the network sampled for the $n$-th MC dropout sample. A high BLAD score indicates a network with high uncertainty about the generated feature matrix. So it tends to be selected to improve the GNN model. Finally, the overall algorithm framework is summarized in Algorithm 1, which shows the optimization of Eq.(4).

---

**Algorithm 1** The framework to train the Generator $Q_\phi$ and the downstream GNN $P_\theta$ using the initial feature matrix $\mathbf{X}$ and the generated feature matrix $\overline{\mathbf{X}}$ selected by the acquisition function

---

**Input:** Adjacency matrix $\mathbf{A}$, feature matrix $\mathbf{X}$

1: Initialize $U$=-inf, $\overline{\mathbf{X}}$, $Q_\phi$, $\overline{\mathbf{X}}'$, and $Q'_\phi$
2: **for** $i = 1$ to the number of generator iterations **do**
3:     Train the generator $Q_\phi$ using $\mathbf{A}$ and $\mathbf{X}$
4:     Generate feature matrix $\overline{\mathbf{X}}$ using $Q_\phi$
5:     Compute $U(\overline{\mathbf{X}})$ using Eq.(10).
6:     **if** $U(\overline{\mathbf{X}}) > U$ **then**
7:         $U = U(\overline{\mathbf{X}})$
8:         **if** $i > N_{warmup}$ **then**
9:             Train GNN $P_\theta$ using $\mathbf{A}$ and $\overline{\mathbf{X}}$ for the number of continued GNN training iterations
10:            $\overline{\mathbf{X}}' = \overline{\mathbf{X}}$,    $Q'_\phi = Q_\phi$
11: $\overline{\mathbf{X}} = \overline{\mathbf{X}}'$,    $Q_\phi = Q'_\phi$
12: Train the downstream GNN $P_\theta$ with the generated feature matrix $\overline{\mathbf{X}}$, and generator $Q_\phi$

---

## 4 DISCUSSION

In this section, we discuss the motivation of this work and provide some analysis.

**Connection to EP-B and GraphSAGE**   We discuss how our proposed model distinguishes from the classical representation learning models on graphs. Previous methods such as EP-B (García-Durán & Niepert, 2017) and GraphSAGE (Hamilton et al., 2017) rely on reconstruction loss function between the central node and its neighbors' embeddings. EP-B aims to minimize the reconstruction error by optimizing the objective $\min \sum_{u \in V \setminus \{v\}} \left[ \gamma + d(\widetilde{\mathbf{X}}_v, \mathbf{X}_v) - d(\widetilde{\mathbf{X}}_v, \mathbf{X}_u) \right]$ where $\mathbf{X}_v$ represents the target node; $\mathbf{X}_u$ denotes the neighbor nodes; $\widetilde{\mathbf{X}}_v = \mathrm{AGG}(\mathbf{X}_l | l \in \mathcal{N}(v))$ indicates the reconstruction from neighbors; and $\gamma$ refers to the bias. Besides, GraphSAGE exploits the negative sampling to differentiate the representations of remote node-pairs. GraphSAGE enforce nearby nodes to have similar representations and to enforce disparate nodes to be distinct by minimizing the objective $\min -E_{u \sim \mathcal{N}(v)} \log \left( (\sigma(\mathbf{X}_u^T \mathbf{X}_v)) \right) - \lambda E_{v_n \sim P_n(v)} \log \left( (\sigma(-\mathbf{X}_{v_n}^T \mathbf{X}_v)) \right)$ where $\mathbf{X}_v$ denotes target node; $\mathbf{X}_u$ represents the neighbor node; $\mathbf{X}_{v_n}$ is disparate node; and $P_n(v)$ is the negative sampling. These approaches build upon the assumption that adjacent nodes share similar attributes. In contrast, our model does not rely on such assumption and instead generates the neighboring node features from the conditional distribution of central node representations. Given the target node, $\mathbf{X}_v$, our aim is to learn the conditional distribution of the neighbor nodes, $\mathbf{X}_u$. A comparison between the reconstruction-based representation learning on graphs and our proposed framework is illustrated in Figure 2. And our local augmentation method is the third paradigm to exploit neighbors in a generative way.

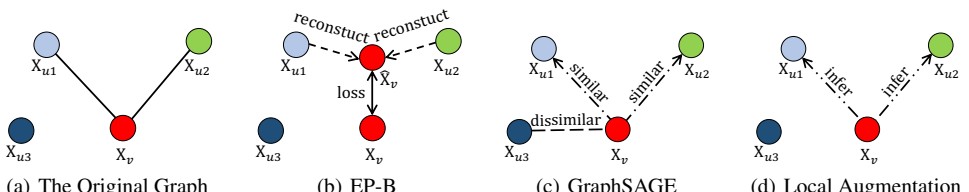

(a) The Original Graph      (b) EP-B      (c) GraphSAGE      (d) Local Augmentation

Figure 2: (a) The original graph. (b) EP-B exploits the neighbors to reconstruct the central node's embedding. (c) GraphSAGE encourages nearby nodes to have similar embeddings. (d) Given the representation of the central node, our aim is to infer the representations of the connected distribution of neighbors.

**Local Augmentation vs. General Augmentation**   General image augmentation algorithms include geometric transformations, feature space augmentation, adversarial training, and generative adversarial networks (Shorten & Khoshgoftaar, 2019). It is impossible to apply geometric transformations directly to graph data augmentation since graphs are sensitive to node permutation. General

adversarial training, feature space augmentation, and generative adversarial networks don't take the graph structure into account. Graphs consist of a set of identities with certain pairs of these identities connected by edges. We need to consider node features and the graph structure when designing the graph data augmentation framework. Our proposed method of local augmentation fully considers these two points. By extracting the neighbors' feature vectors, we have enough data points to learn the distribution. There are two benefits to designing local augmentation. First, by taking the sub-graph structure and feature representation associated with this sub-graph structure as input for the generative model, we can learn the information of the sub-graph structure. Second, the number of data points to learn the distribution depends on the node degree. This assures that we have enough data points compared with the general feature augmentation and we can learn a better distribution.

**Complementing missing information** Jia & Benson (2020) points out that some attribute information might be missing on a subset of vertices. By learning the distribution of node representations from the observed data, we can utilize the produced node representations from the generative model to complement the information missing in the nodes' attributes, which boosts the robustness of downstream tasks. And we show that our model still works in the scenario that nodes lose a certain percentage of attributes. In other words, we can exploit the well-learned distribution to complement the contextual information of the local neighborhood to enhance the locality of the node representations.

## 5 EXPERIMENTS

In this section, we evaluate the performance of our proposed model on semi-supervised node classification tasks on a variety of public graph datasets and compare our model with the state-of-the-art graph neural networks. We also carry out additional experiments to showcase the necessity of our design and its robustness to missing information.

### 5.1 DATASETS

We utilize seven public graph datasets (Cora, Citeseer, Pubmed, Squirrel, Actor, Chameleon, and Cornell) for semi-supervised node classification tasks. The details of these datasets can be found in the appendix.

Table 2: Classification results on fixed split (%)

| Method | Cora | Citeseer | Pubmed |
|---|---|---|---|
| Chebyshev (Defferrard et al., 2016) | 81.2 | 69.8 | 74.4 |
| APPNP (Klicpera et al., 2019) | 83.8 | 71.6 | 79.7 |
| MixHop (Abu-El-Haija et al., 2019) | 81.9 | 71.4 | 80.8 |
| Graph U-net (Gao & Ji, 2019) | 84.4 | 73.2 | 79.6 |
| GSNN-M (Wang et al., 2020a) | 83.9 | 72.2 | 79.1 |
| $S^2$GC (Zhu & Koniusz, 2021) | 83.5 | 73.6 | 80.2 |
| GCN (Kipf & Welling, 2017) | 81.6 | 70.3 | 78.9 |
| G-GCN (Zhu et al., 2020) | 83.7 | 71.3 | 80.9 |
| DropEdge-GCN (Rong et al., 2020) | 82.8 | 72.3 | 79.6 |
| GAUG-O-GCN (Zhao et al., 2021) | 83.6 | 73.3 | 79.3 |
| LA-GCN | 84.1 | 72.5 | 81.3 |
| GAT (Veličković et al., 2018) | 83.0 | 70.4 | OOM |
| LA-GAT | 83.9 | 72.3 | OOM |
| GCNII (Chen et al., 2020) | 85.2 | 73.1 | 80.0 |
| LA-GCNII | 85.2 | 73.7 | 81.6 |
| GRAND (Feng et al., 2020) | 85.4 | 75.4 | 82.7 |
| LA-GRAND | **85.8** | **75.8** | **83.3** |

### 5.2 SEMI-SUPERVISED NODE CLASSIFICATION

**Baselines and Experimental Setup.** We apply the standard fixed splits (Yang et al., 2016) on three datasets Cora, Citeseer, and Pubmed, with 20 nodes per class for training, 500 nodes for validation, and 1,000 nodes for testing. And we consider four backbones: GCN (Kipf & Welling, 2017), GAT (Veličković et al., 2018), GCNII (Chen et al., 2020), and GRAND (Feng et al., 2020) to evaluate our proposed framework and compare our model against state-of-the-art models including 1) backbone models: Chebyshev (Defferrard et al., 2016), GCN, GAT,

Table 3: Classification results on random split (%)

| Method | Squirrel | Actor | Chameleon | Cornell |
|---|---|---|---|---|
| APPNP | 21.6 | 32.1 | 33.0 | **58.7** |
| $S^2$GC | 21.3 | 27.8 | 30.2 | 57.2 |
| GCN | 22.5 | 26.2 | 25.1 | 55.7 |
| DropEdge-GCN | 21.9 | 26.5 | 25.0 | 53.6 |
| LA-GCN | 23.2 | 27.0 | 28.9 | 56.1 |
| GAT | 24.2 | 27.2 | 34.8 | 55.8 |
| LA-GAT | 28.2 | 27.4 | **38.6** | 56.5 |
| GCNII | 25.3 | 31.9 | 30.2 | 57.3 |
| LA-GCNII | **28.6** | **32.7** | 32.5 | 56.6 |

APPNP (Klicpera et al., 2019), Graph U-net (Gao & Ji, 2019), MixHop (Abu-El-Haija et al., 2019), GCNII, GSNN-M (Wang et al., 2020a), $S^2$GC (Zhu & Koniusz, 2021), and GRAND and 2) feature-level and topology-level augmentation models: G-GNNs (Zhu et al., 2020), DropEdge (Rong et al.,

2020) and GAUG-O (Zhao et al., 2021). For four datasets Squirrel, Actor, Chameleon, and Cornell, we take 10 random splits (Shchur et al., 2018) where 10%, 30%, and 60% of the date for training, validation, testing; measure the performance of GCN, GAT, GCNII, and corresponding modified models.

**Results**   For three datasets Cora, Citeseer, and Pubmed, we report the mean classification accuracy on the test nodes of all our models after 100 runs and report the values after running the experiments of their models with our server under their setting hyperparameters in their original papers. The results of the evaluation experiments are summarized in Tables 2, 3, and in the appendix, which demonstrate that the backbone models equipped with our method achieve the best performance across all the datasets except the Cornell dataset. More specifically, we can improve upon GCN by a margin of 2.5%, 2.2%, and 2.4% on Cora, Citeseer, and Pubmed respectively. Moreover, LA-GNN outperforms other backbone models including GAT and GCNII as well as data augmentation models (Zhu et al., 2020; Rong et al., 2020; Zhao et al., 2021) on these citation network datasets. Besids, we also provide the analysis of the distribution of our generated feature matrix. And Figure 3 shows the distribution of the attributes of the original and inference neighbors, which can demonstrate our inference feature matrix follow the distribution of the initial feature matrix.

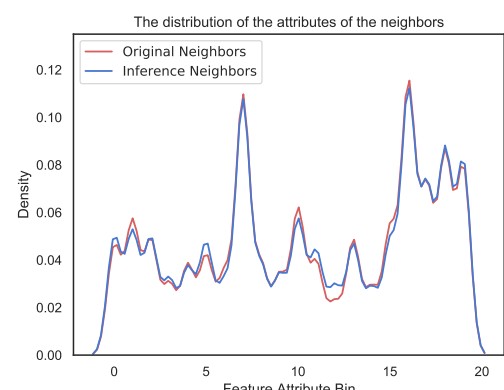

Figure 3: The distribution of the attribute bin of the inference neighbors vs. the distribution of the attribute bin of the original neighbors, with KL divergence = 0.0026. The value of each feature bin is the sum of the attribute values of multiple dimensions of the feature vector. We split the feature vector into multiple feature bins.

## 5.3   ABLATION STUDY

In this section, to demonstrate the effectiveness of our proposed generative framework, we conduct experiments that compare LA-GNN to several of its ablated variants without generative modeling. The results are shown in Table 4. "GCN + width" only increases the first network layer width for GCN and GCNII to match LA-GNN without giving generated samples as input. "+ concatenation" only replaces the generated feature matrix of LA-GNN with the original feature matrix of the central node. "+ plain neighborhood" replaces the generated feature matrix of LA-GNN with a neighborhood feature matrix where each row corresponds to the feature vector of the randomly sampled neighbor. The

Table 4: Effects of different components of our framework evaluated on the standard split of the Cora, Citeseer and Pubmed dataset.

| Method | Cora | Citeseer | Pubmed |
|---|---|---|---|
| GCN | 81.6 | 70.3 | 78.9 |
| GCNII | 85.2 | 73.1 | 80.0 |
| GCN + width | 82.0 | 71.4 | 79.5 |
| GCN + concatenation | 81.8 | 71.6 | 78.8 |
| GCN + plain neighborhood | 80.9 | 68.8 | 75.0 |
| GCNII + width | 85.1 | 73.1 | 80.2 |
| GCNII + concatenation | 85.2 | 73.3 | 80.2 |
| GCNII + plain neighborhood | 83.3 | 71.9 | 78.1 |
| LA-GCN | 84.1 | 72.5 | 81.3 |
| LA-GCNII | 85.2 | 73.7 | 81.6 |

results show that the first two variants provide no notable improvement for the backbone models, and the third variant even results in degradation. By eliminating the possibility that these confounding factors irrelevant to our core approach may contribute to the final performance, it's evident that the performance gain in Table 2 and 3 are due to our proposed generative local augmentation framework.

## 5.4   ROBUSTNESS TO MISSING INFORMATION

In this section, we conduct experiments to verify that our proposed framework can robustify downstream tasks against missing information in the feature attributes. Specifically, we mask a certain percentage of the attributes of each feature vector and use the same pipeline to do augmentation for the masked feature matrix. As shown in Table 5, we can see that as the mask ratio increases,

the gap of the performance between the GCN and LA-GCN enlarges in most cases in Cora and Citeseer, which corroborates our insight discussed in Section 4. Since there exists large redundancy in the features of the Pubmed dataset, the performance of GCN and LA-GCN decreases little as the mask ratio increases and the gap of the performance does not enlarge. To conclude, our model can complement the contextual information of the local neighborhood to enhance the locality of the node representations.

Table 5: Summary of results on recovering study in terms of classification accuracy (%). ↓ means a decrease compared with the accuracy if features are not masked.

| Dataset | Cora | | | | Citeseer | | | | Pubmed | | | |
|---|---|---|---|---|---|---|---|---|---|---|---|---|
| Mask Ratio | 0.1 | 0.2 | 0.4 | 0.8 | 0.1 | 0.2 | 0.4 | 0.8 | 0.1 | 0.2 | 0.4 | 0.8 |
| GCN | 81.0 (↓0.6) | 80.6 (↓1.0) | 80.1 (↓1.5) | 76.0 (↓5.6) | 70.1 (↓0.2) | 69.3 (↓1.0) | 67.2 (↓3.1) | 61.0 (↓9.3) | 78.5 (↓0.4) | 78.5 (↓0.4) | 77.5 (↓1.4) | 76.9 (↓2.0) |
| LA-GCN | 83.5 (↓0.6) | 83.1 (↓1.0) | 81.6 (↓2.5) | 81.1 (↓3.0) | 72.2 (↓0.3) | 71.7 (↓0.8) | 69.3 (↓3.2) | 65.9 (↓6.6) | 81.4 (↓0.1) | 80.9 (↓0.6) | 80.5 (↓1.0) | 79.4 (↓2.1) |

## 6 RELATED WORK

**Graph Neural Networks** In general, convolution in the graph domain involves non-spectral (spatial) and spectral approaches. Non-spectral methods generalize convolutions operating on spatially close neighbors to the graph domain, such as Duvenaud et al. (2015); Atwood & Towsley (2016); Niepert et al. (2016); Monti et al. (2017). Spectral approaches define the convolution operations based on the spectral formulation, such as Bruna et al. (2014); Defferrard et al. (2016); Kipf & Welling (2017). Recently, several methods (Abu-El-Haija et al., 2019; Liao et al., 2019) based on GCN have been proposed to obtain the higher-order filters. Besides, GAT (Veličković et al., 2018), Graph U-Nets (Gao & Ji, 2019) combine attention networks and pooling operation with GNN separately, which achieve state-of-the-art performance on node and link classification tasks. In this work, local augmentation can be applied on various backbone models to improve performance.

**Graph Generative Models** Generative models (Goodfellow et al., 2014; Kingma & Welling, 2013) are powerful tools of learning data distribution through unsupervised learning, and they have achieved tremendous success in various applications. Recently, researchers have proposed several interesting generative models for graph data generation. Variational graph auto-encoder (VGAE) (Kipf & Welling, 2016) makes use of latent variables and learns interpretable latent representations for undirected graphs. Salha et al. (2019) replace the GCN encoder in VGAE with a simple linear model and emphasize the effectiveness of a simple node encoding scheme. Xu et al. (2019) propose a generative model framework to learn node representations, by sampling graph generation sequences constructed from observed graph data. ConDgen (Yang et al., 2019) exploits the GCN encoder to handle the inherent challenges of flexible context-structure conditioning and permutation-invariant generation. Besides, some methods have been proposed to apply the graph generative models in various applications such as graph matching (Simonovsky & Komodakis, 2018), molecule design (Liu et al., 2018), retrosynthesis prediction (Shi et al., 2020) and chemical design (Samanta et al., 2018). Compared with these approaches mainly focusing on structure generation, our model takes full use of the power of the generative model for feature representation generation, which can serve as an enhanced technique for the downstream backbone models.

## 7 CONCLUSION

We propose local augmentation, a brand-new technique that exploits the generative model to learn the conditional distribution of the central node's neighbors' feature representations given its representation. We can augment more 1-hop neighbors from a well-trained generative model to enhance the performance of backbone GNN models. Experiments show that our model can improve performance across various GNN architectures and benchmark datasets by enriching local information. Besides, our model achieves new state-of-the-art results on various semi-supervised node classification tasks. One limitation of our proposed framework is that we do not exploit the 2-hop neighbors or use the random walk to find more related neighbors for the central node. And one future work is that we can extract more 2/3-hop neighbors if the central node's degree is small and learn the conditional distribution for random sampling nodes if the graph is large.

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

## A    PROOF OF EQ.(6)

We give more details of the derivation of the generator ELBO as follows:

$$
\begin{aligned}
\log p_\psi(\mathbf{X}_j|\mathbf{X}_i) &= \int q_\varphi(\mathbf{z}|\mathbf{X}_j,\mathbf{X}_i)\log p_\psi(\mathbf{X}_i|\mathbf{X}_i)\mathrm{d}\mathbf{z} \\
&= \int q_\varphi(\mathbf{z}|\mathbf{X}_j,\mathbf{X}_i)\log\frac{p_\psi(\mathbf{X}_j,\mathbf{X}_i)}{p_\psi(\mathbf{X}_i)}\mathrm{d}\mathbf{z} \\
&= \int q_\varphi(\mathbf{z}|\mathbf{X}_j,\mathbf{X}_i)\log\frac{p_\psi(\mathbf{X}_j,\mathbf{X}_i)p_\psi(\mathbf{X}_i,\mathbf{X}_i,\mathbf{z})}{p_\psi(\mathbf{X}_i)p_\psi(\mathbf{X}_i,\mathbf{X}_i,\mathbf{z})}\mathrm{d}\mathbf{z} \\
&= \int q_\varphi(\mathbf{z}|\mathbf{X}_i,\mathbf{X}_i)\log\frac{p_\psi(\mathbf{X}_j,\mathbf{X}_i,\mathbf{z})}{p_\psi(\mathbf{X}_i)}\frac{1}{\frac{p_\psi(\mathbf{X}_j,\mathbf{X}_i,\mathbf{z})}{p_\psi(\mathbf{X}_j,\mathbf{X}_i)}}\mathrm{d}\mathbf{z} \\
&= \int q_\varphi(\mathbf{z}|\mathbf{X}_j,\mathbf{X}_i)\log\frac{p_\psi(\mathbf{X}_j,\mathbf{z}|\mathbf{X}_i)}{p_\psi(\mathbf{z}|\mathbf{X}_j,\mathbf{X}_i)}\mathrm{d}\mathbf{z} \\
&= \int q_\varphi(\mathbf{z}|\mathbf{X}_j,\mathbf{X}_i)\log\frac{p_\psi(\mathbf{X}_j,\mathbf{z}|\mathbf{X}_i)}{p_\psi(\mathbf{z}|\mathbf{X}_j,\mathbf{X}_i)}\frac{q_\varphi(\mathbf{z}|\mathbf{X}_j,\mathbf{X}_i)}{q_\varphi(\mathbf{z}|\mathbf{X}_j,\mathbf{X}_i)}\mathrm{d}\mathbf{z} \\
&= \int q_\varphi(\mathbf{z}|\mathbf{X}_j,\mathbf{X}_i)\left(\log\frac{p_\psi(\mathbf{X}_j,\mathbf{z}|\mathbf{X}_i)}{q_\varphi(\mathbf{z}|\mathbf{X}_j,\mathbf{X}_i)}+\log\frac{q_\varphi(\mathbf{z}|\mathbf{X}_j,\mathbf{X}_i)}{p_\psi(\mathbf{z}|\mathbf{X}_j,\mathbf{X}_i)}\right)\mathrm{d}\mathbf{z} \\
&= \int q_\varphi(\mathbf{z}|\mathbf{X}_j,\mathbf{X}_i)\log\frac{p_\psi(\mathbf{X}_j,\mathbf{z}|\mathbf{X}_i)}{q_\varphi(\mathbf{z}|\mathbf{X}_j,\mathbf{X}_i)}\mathrm{d}\mathbf{z} + KL(q_\varphi(\mathbf{z}|\mathbf{X}_j,\mathbf{X}_i)||p_\psi(\mathbf{z}|\mathbf{X}_j,\mathbf{X}_i)) \\
&\geq \int q_\varphi(\mathbf{z}|\mathbf{X}_j,\mathbf{X}_i)\log\frac{p_\psi(\mathbf{X}_j,\mathbf{z}|\mathbf{X}_i)}{q_\varphi(\mathbf{z}|\mathbf{X}_j,\mathbf{X}_i)}\mathrm{d}\mathbf{z}
\end{aligned}
$$

$$
\begin{aligned}
L_{ELBO} &= \int q_\varphi(\mathbf{z}|\mathbf{X}_j,\mathbf{X}_i)\log\frac{p_\psi(\mathbf{X}_j,\mathbf{z}|\mathbf{X}_i)}{q_\varphi(\mathbf{z}|\mathbf{X}_j,\mathbf{X}_i)}\mathrm{d}\mathbf{z} \\
&= \int q_\varphi(\mathbf{z}|\mathbf{X}_j,\mathbf{X}_i)\log\frac{p_\psi(\mathbf{X}_j,\mathbf{X}_i,\mathbf{z})}{q_\varphi(\mathbf{z}|\mathbf{X}_j,\mathbf{X}_i)p_\psi(\mathbf{X}_i)}\mathrm{d}\mathbf{z} \\
&= \int q_\varphi(\mathbf{z}|\mathbf{X}_j,\mathbf{X}_i)\log\frac{p_\psi(\mathbf{X}_j|\mathbf{X}_i,\mathbf{z})p_\psi(\mathbf{X}_i,\mathbf{z})}{q_\varphi(\mathbf{z}|\mathbf{X}_j,\mathbf{X}_i)p_\psi(\mathbf{X}_i)}\mathrm{d}\mathbf{z} \\
&= \int q_\varphi(\mathbf{z}|\mathbf{X}_j,\mathbf{X}_i)\log\frac{p_\psi(\mathbf{X}_j|\mathbf{X}_i,\mathbf{z})p_\psi(\mathbf{z}|\mathbf{X}_i)}{q_\varphi(\mathbf{z}|\mathbf{X}_j,\mathbf{X}_i)}\mathrm{d}\mathbf{z} \\
&= \int q_\varphi(\mathbf{z}|\mathbf{X}_j,\mathbf{X}_i)\log\frac{p_\psi(\mathbf{z}|\mathbf{X}_i)}{q_\varphi(\mathbf{z}|\mathbf{X}_j,\mathbf{X}_i)}\mathrm{d}\mathbf{z} + \int q_\varphi(\mathbf{z}|\mathbf{X}_j,\mathbf{X}_i)\log p_\psi(\mathbf{X}_j|\mathbf{X}_i,\mathbf{z})\mathrm{d}\mathbf{z} \\
&= -KL(q_\varphi(\mathbf{z}|\mathbf{X}_j,\mathbf{X}_i)||p_\psi(\mathbf{z}|\mathbf{X}_i)) + \int q_\varphi(\mathbf{z}|\mathbf{X}_j,\mathbf{X}_i)\log p_\psi(\mathbf{X}_j|\mathbf{X}_i,\mathbf{z})\mathrm{d}\mathbf{z}
\end{aligned}
$$

# B    REPRODUCIBILITY

## B.1    DATASETS DETAILS

Cora, Citeseer, and Pubmed are standard citation network benchmark datasets Sen et al. (2008). In these datasets, nodes represent documents, and edges denote citations; node feature corresponds to elements of a bag-of-words representation of a document, and node label corresponds to one of the academic topics. Besides, we utilize four datasets used in Pei et al. (2020) for evaluation. Chameleon and squirrel are two page-page networks on specific topics in Wikipedia Rozemberczki et al. (2021). In these datasets, nodes represent web pages, and edges denote mutual links between pages; node features correspond to several informative nouns in the Wikipedia pages and labels correspond to the number of the average monthly traffic of the web page. WebKB[1] is a webpage dataset collected from various universities. We use the one subdataset of it, Cornell. In this dataset, nodes represent web pages, and edges are hyperlinks between them; node features correspond to the bag-of-words representation of web pages and labels correspond to five categories, student, project, course, staff, and faculty. Film dataset is the actor-only induced subgraph of the film-directoractor-writer network Tang et al. (2009). In this dataset, Nodes represent actors, and edges denote co-occurrence on the same Wikipedia page; node features correspond to some keywords in the Wikipedia pages and labels correspond to five categories in terms of words of actor's Wikipedia. All the dataset statistics are summarized in Table 6.

Table 6: Datasets statistics

| Dataset | Cora | Cite. | Pubm. | Cham. | Squi. | Actor | Corn. |
|---------|------|-------|-------|-------|-------|-------|-------|
| # Nodes | 2708 | 3327 | 19717 | 2277 | 5201 | 7600 | 183 |
| # Edges | 5429 | 4732 | 44338 | 36101 | 217073 | 33544 | 295 |
| # Features | 1433 | 3703 | 500 | 2325 | 2089 | 931 | 1703 |
| # Classes | 7 | 6 | 3 | 5 | 5 | 5 | 5 |

## B.2    IMPLEMENTATION DETAILS

We use Pytorch (Paszke et al., 2019) to implement LA-GNNs. The codes of $S^2$GC (Zhu & Koniusz, 2021), LA-GCN, LA-GAT, LA-GCNII, LA-GRAND, and DropEdge-GCN are implemented referring to Pytorch implementation of $S^2$GC[2], GCN[3] (Kipf & Welling, 2017), GAT[4] (Veličković et al., 2018), GCNII[5] (Chen et al., 2020) GRAND[6] (Feng et al., 2020), and DropEdge-GCN[7] (Rong et al., 2020). Besides, we implement APPNP (Klicpera et al., 2019) with DGL (Wang et al., 2019) version of APPNP[8]. The datasets Cora, Citeseer, Pubmed are downloaded from TensorFlow (Abadi et al., 2016) implementation of GCN[9], and the datasets Chameleon, Squirrel, Actor, and Cornell are downloaded from the implementation of Geom-GCN[10](Pei et al., 2020). All the experiments in this work are conducted on a single NVIDIA Tesla V100 with 32GB memory size. The operating system behind the Docker where the experiments are running is Red Hat 4.8.2-16. And the software that we use for experiments are Python 3.6.8, numpy 1.19.2, sklearn 0.0, scipy 1.5.4, networkx 2.5.1, torch 1.6.0, torchvision 0.7.0, CUDA 10.2.89, and CUDNN 8.0.2.

---

[1]http://www.cs.cmu.edu/afs/cs.cmu.edu/project/theo-11/www/wwkb

[2]https://github.com/allenhaozhu/SSGC

[3]https://github.com/tkipf/pygcn

[4]https://github.com/Diego999/pyGAT

[5]https://github.com/chennnM/GCNII

[6]https://github.com/THUDM/GRAND

[7]https://github.com/DropEdge/DropEdge

[8]https://github.com/dmlc/dgl/tree/master/examples/pytorch/appnp

[9]https://github.com/tkipf/gcn/tree/master/gcn/data

[10]https://github.com/graphdml-uiuc-jlu/geom-gcn/tree/master/new_data

## B.3 HYPERPARAMETER DETAILS

LA-GNNs introduce an additional parameter, that is the hidden layer for generated feature matrix $\overline{\mathbf{X}}$ before concatenation. The difference of architectures between GCN and LA-GCN can be found in Figure 4, and the LA-GCNII architecture can be found in Figure 5.

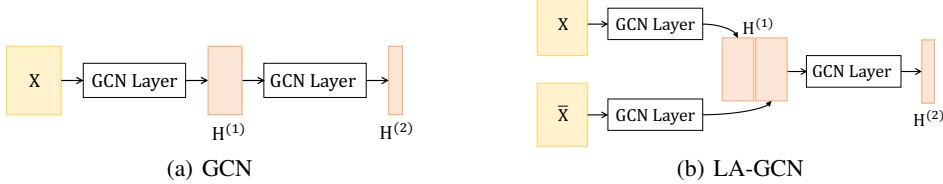

(a) GCN

(b) LA-GCN

Figure 4: GCN and LA-GCN architectures. The difference between GCN and LA-GCN architectures is that the LA-GCN has an additional convolutional layer for $\overline{\mathbf{X}}$ and it uses a concatenation operation to mix the hidden representations.

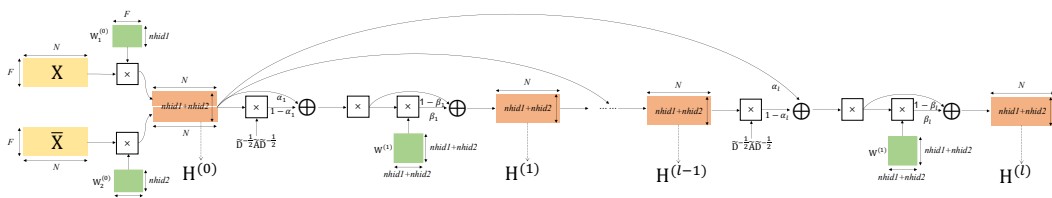

Figure 5: LA-GCNII architecture. The difference between GCNII and LA-GCNII is that the LA-GCNII has an additional MLP layer for $\overline{\mathbf{X}}$ and it uses a concatenation operation to mix the hidden representations.

The difference of hyperparameters between the GCN and LA-GCN is only the hidden layer size before concatenation. For the LA-GCNII, LA-GAT, LA-GRAND, we tune the hyperparameters in the same way as described in their original papers with validation set.

