# OpenReview forum: "Local Augmentation for Graph Neural Networks"
_ICLR.cc/2022/Conference — ICLR 2022 Submitted_

### Official Review · Reviewer_E5PW · 2021-10-30

**Correctness:** 2
**Technical Novelty And Significance:** 3
**Empirical Novelty And Significance:** 1
**Recommendation:** 3
**Confidence:** 4

**Main Review:**

Strengths
- This paper solves an important problem, graph augmentation, which has a large impact on many graph-related tasks such as node classification.
- Eq. (2) to (4) provide a good motivation for why augmentation is able to improve the performance of a classifier, in a probabilistic view.
- The proposed approach improves various GNN models in benchmark datasets.

Weaknesses
- The node index i just appears at Eq. (5) without any background information. Does i iterate over all nodes or only the training nodes? Do we assume the independence between different i values? Clarification is needed.
- The discussion before the “Optimization of the MLE” paragraph seems redundant. The local augmentation is not very different from previous approaches, since X_v works as the condition variable of the variational inference to infer the X_u1 and X_u2; X_v is expected to be similar to X_u1 and X_u2 for better estimation.
- Eq. (6) is incomplete and needs much clarification. a) What are the structures of networks f and g? b) Do the authors use the reparameterization trick for generating Z? c) Why do the authors use the same f function for both the encoder and decoder functions? d) What is the meaning of running variational inference if all neighbors of each node have Z sampled from the same distribution N(0, I)?
- The name of Section 2.3 should be active learning rather than importance sampling, since a) the goal is not to estimate an unknown probability distribution, and b) the augmentation is done for improving only the current status of the classifier; there is no guarantee that the quality of \bar{X} improves globally.
- The proposed LA-GNN structure contains much more parameters, which result in a higher risk of overfitting as a consequence. An additional study is needed for the number of parameters, running time, and the sensitivity to hyperparameters.
- It is unclear what the authors want to say with Figure 3; what does it mean?
- The experiment of Section 4.3 seems meaningless, since the approaches with partial ideas such as “+width” or “+concatenation” are not reasonable at all. For example, “+concatenation” just repeats the original features multiple times. Such results do not give any insights on the proposed approach.


**Summary Of The Paper:**

Data augmentation remains under-explored on graph-structured data. In this work, the authors introduce local augmentation that enhances the feature of each node by its local subgraph structure. Specifically, they model the data augmentation as a feature generation process: given a node’s feature, they learn the conditional distribution of its neighbors’ features to make a generative augmentation model. Experiments show that the proposed local augmentation makes performance improvement on real-world data.

**Summary Of The Review:**

This paper proposes a new graph augmentation method that enhances the performance of various GNN models by augmenting input features. However, the technical quality of this paper is insufficient, and the ideas, writings, and experiments are unclear and overwrapping. The idea of running variational inference for graph augmentation is intriguing, so I suggest the authors further develop their approach.

---

> ### Author Response · Authors · 2021-11-16
> **Response to Reviewer E5PW (Part 1)**
>
> We thank the reviewer for recognition of several key elements of our paper:
> The key contribution/novelty: performing feature-level augmentation on graph by learning a distribution conditioned on local structure and the center node's features, via a generative model.
>
> **Please refer to our general response about the intuitions and contributions of our paper.**
>
> First of all, our main contribution is not running variational inference for graph augmentation, we just use CVAE for generation. Our main contribution is to enhance the locality of node representations using local augmentation. Besides, we did omit some details of CVAE, because we simply employ the conventional CVAE in our paper.
>
> **Q1: The node index i just appears at Eq. (5) without any background information. Does i iterate over all nodes or only the training nodes? Do we assume the independence between different i values? Clarification is needed.**
>
>
>
> A1: I'm very sorry for that. I iterates over all nodes. For Eq. (5), there is an error of the notation. Our $p_{\psi}\left(X_{j} \mid X_{i}\right)$ means $p\left(X_{j} \mid X_{i}, j \in N_i\right)$. For $X_{j}$ and $X_{i}$, especially for those connected samples in the graph, they are NOT independent. Our algorithm intends to learn the first order (graph) conditional distribution.
>
>
>
>
>
> **Q2: The discussion before the “Optimization of the MLE” paragraph seems redundant. The local augmentation is not very different from previous approaches, since X_v works as the condition variable of the variational inference to infer the X_u1 and X_u2; X_v is expected to be similar to X_u1 and X_u2 for better estimation.**
>
>
>
> A2: We have placed this part in our newly added section "Discussion". First of all, EP-B and GraphSAGE have no relationship with variational inference. Secondly, they are two traditional methods to learn graph embedding. As I have discussed in this part, GraphSAGE enforces nearby nodes to have similar representations and to enforce disparate nodes to be distinct, and EP-B exploits the neighbors to reconstruct the central node’s embedding. Thirdly, our method does not rely on the assumption that adjacent nodes share similar embeddings. And we do not expect that $X_v$ is similar to $X_{u1}$ and $X_{u2}$. As you can see from the experiment results in table 3, the accuracies of Squirrel, Actor, Chameleon, and Cornell are much smaller than Cora, Citeseer, and Pubmed although they have larger proportions of the training set than Cora, Citeseer, and Pubmed. In other words, Squirrel, Actor, Chameleon, and Cornell do not have sufficient homophily properties [2]. But our method still works. For one reason, our augmentation method has the same effect as general augmentation in Computer Vision and Natural Language Processing. For another reason, our augmentation method is just to learn the conditional distribution of the connected nodes and we do not rely on the assumption that connected neighbors have similar feature representations. Last, our method is the third paradigm to exploit neighbors. What's more, our method is not based on/for graph embedding. We just exploit the well-learned conditional distribution for local augmentation to enhance the locality of node representations.
>
>
>
>
>
> **Q3: Eq. (6) is incomplete and needs much clarification. a) What are the structures of networks f and g? b) Do the authors use the reparameterization trick for generating Z? c) Why do the authors use the same f function for both the encoder and decoder functions? d) What is the meaning of running variational inference if all neighbors of each node have Z sampled from the same distribution N(0, I)?**
>
>
>
> A3: We are very sorry that we didn't make enough clarification about Eq.(6) and we make an error in the notation. a) f and g are parts of encoder, the encoder is a two-layer MLP. f and g share the first layer, and their second layers use different parameters. f is for learning means, g is for learning variances. b) We use the reparameterization trick for generating Z. c) We do not use the same f function for both the encoder and decoder functions. f and g consitute the encoder. Decoder is not represented by any notation. And we will use h to represent decoder. d) No, we don't sample Z from the same distribution. We first get means and variances, and then use reparameterization trick to get the latent variable Z. Different neighbor can get different means and variances. And we have added some clarification about Eq.(6) based on your suggestion. Please see our new submission of the paper.
>
>
>
>
>
> **Q4: The name of Section 2.3 should be active learning rather than importance sampling, since a) the goal is not to estimate an unknown probability distribution, and b) the augmentation is done for improving only the current status of the classifier; there is no guarantee that the quality of \bar{X} improves globally.**
>
>
>
> A4: We have renamed section 2.3 based on your suggestion. Please see our new submission of the paper.

---

> > ### Author Response · Authors · 2021-11-16
> > **Response to Reviewer E5PW (Part 2)**
> >
> > **Q5: The proposed LA-GNN structure contains much more parameters, which result in a higher risk of overfitting as a consequence. An additional study is needed for the number of parameters, running time, and the sensitivity to hyperparameters.**
> >
> >
> >
> > A5: As you can see from the experiment results in Tables 2 and 3, we improve the performance over various GNN models. In fact, our proposed LA-GNN structure only exploits the generated feature matrix, and even LA-GRAND and LA-GAT do have the same architecture as GRAND and GAT. Besides, our method does not overfit since we provide enough generated feature matrices for the input of LA-GNN. For LA-GCNII and LA-GCN, we can add additional parameters only in the first layer, and the parameters of other layers are the same as GCNII and GCN. Besides, we increase the number of parameters of LA-GCN and LA-GCNII over the corresponding backbones to just exploit the generated feature matrix. But the time complexity is of the same order of magnitude with corresponding backbones. For the sensitivity to hyperparameters, And the proposed LA-GNN structure is robust to hyperparameters. We can improve performance on all hyperparameters. And we select the hyperparameters based on the performance of the validation set. What's more, other graph data augmentation methods also introduce additional parameters. I think the key to introducing additional parameters is whether this performance improvement is worth adding more parameters. In conclusion, our method does not introduce too many parameters, and the training time is the same order of magnitude as the corresponding backbone. And we do not introduce additional parameters for GRAND. More importantly, our method can improve performance very much, and even achieve the latest SOTA performance.
> >
> >
> >
> >
> >
> > **Q6: It is unclear what the authors want to say with Figure 3; what does it mean?**
> >
> >
> >
> > A6: It shows that we successfully learned the conditional distribution, which can demonstrate our inference feature matrix follows the distribution of the initial feature matrix.
> >
> >
> >
> >
> >
> > **Q7: The experiment of Section 4.3 seems meaningless, since the approaches with partial ideas such as “+width” or “+concatenation” are not reasonable at all. For example, “+concatenation” just repeats the original features multiple times. Such results do not give any insights on the proposed approach.**
> >
> >
> >
> > A7: This ablation study just shows that the generated feature matrix can improve performance. And the performance improvement is not due to the additional parameters.

---

> > > ### Author Response · Authors · 2021-11-25
> > > **Response to Reviewer E5PW (Part 3)**
> > >
> > > We think that the focus of your review is that we use variational inference for graph data augmentation. We have some problems with the method description, but according to your suggestion, we have solved the problem. What we want to say is that cvae and variational inference are just implementations, and our motivation and intuition for local augmentation are more important. In 2019 and 2020, GNN researchers mostly explored the over smoothing issue of GNN. How to train a deep GNN is the focus of research in the last two years. But when GCNII [1] successfully trained a deep model, they also added a residual connection to the input layer. They also emphasized the importance of the locality of node representations in the paper. In other words, local information is the most important for training GNN. Recently, GNN researchers are exploring how to train a good GNN with local sub-structure information. Zeng et al. (2021) [3] pointed out that the key for GNN is to smooth the local neighborhood into informative representation, no matter how deep it is. And they decouple the depth and scope of GNNs to help capture local graph structure. Besides, Zhang et al. (2021) [4] and a submission to ICLR [5] utilize local subgraph structure to learn good representations. Our method explores how to use local information to learn good node representations from another perspective.  Our motivation is that one property of the graph is that the number of nodes in the local neighborhood is far fewer than higher-order neighbors. And this property restricts the expressive power of GNNs due to the limited neighbors in the local structure. A very intuitive idea is to use data augmentation to increase the number of nodes in the local substructure. The generative model is just implementation and is not our main contribution. We sincerely hope that our work can contribute to the GNN community. If you have any concerns, please let us know. We are very happy to answer your any question.
> > >
> > >
> > > [1] Ming Chen, Zhewei Wei, Zengfeng Huang, Bolin Ding, and Yaliang Li. Simple and deep graph convolutional networks. In International Conference on Machine Learning, pp. 1725–1735. PMLR, 2020.
> > >
> > > [2] Jiong Zhu, Yujun Yan, Lingxiao Zhao, Mark Heimann, Leman Akoglu, Danai Koutra. Beyond Homophily in Graph Neural Networks: Current Limitations and Effective Designs. In Advances in Neural Information Processing Systems, 2020.
> > >
> > > [3] Hanqing Zeng, Muhan Zhang, Yinglong Xia, Ajitesh Srivastava, Andrey Malevich, Rajgopal Kannan, Viktor Prasanna, Long Jin, and Ren Chen. Decoupling the depth and scope of graph neural networks. In Advances in Neural Information Processing Systems, 2021.
> > >
> > > [4] Muhan Zhang, Pan Li. Nested Graph Neural Networks. In Advances in Neural Information Processing Systems, 2021.
> > >
> > > [5] https://openreview.net/forum?id=uxgg9o7bI_3

---

### Official Review · Reviewer_cVVh · 2021-11-02

**Correctness:** 3
**Technical Novelty And Significance:** 4
**Empirical Novelty And Significance:** 3
**Recommendation:** 5
**Confidence:** 3

**Main Review:**

This paper presents an interesting method to enhance the graph representations with the proposed local augmentation techniques. The authors propose to leveraging a generator to vary the node features conditioned on the neighbor information in the subgraph.  The generated samples are further re-selected with an importance sampling technique to ensure the generated features are informative to enhance representation learning.

From the side of technique in the paper, the proposed method well differs the current augmentation methods and shows the improvement on different backbones. The presentation of the technical part is generally well written and clear. However, I still have some small confusion about the notations in section 2.2. For example: Is $p_\psi(z|X_i)$ a parameterized by $\psi$ or not? In the text I found $p_\psi(z|X_i) = N(0,I)$.


The experiments show the results of classification with different backbones on different datasets and we can see the improvement of local augmentations. Some expected ablations are not shown; thus I have some suggestions for this part to improve this paper:

- I guess it is non-trivial to re-organize Table 3, as it would be easier to compare if the authors could put methods on the same backbone together (e.g. GCN, G-GCN, DE-GCN, GAUG-OGCN and LA-GCN).

- If it is possible, it will be better to show the results of GAT, GCNII and GRAND with the other augmentation methods.

- I am curious how the proposed local augmentation performs when combined with other augmentation techniques. For example, will it further improve the GNNs by putting LA and DropEdge together?

- How does the importance sampling module improve the generated feature quality? Probably it could be shown in Fig 3 to justify the importance of the proposed module.

------- update after rebuttal ---------------


After carefully reading all the reviews, the authors' responses, and corresponding revisions, I think the quality of this paper is better than the original submission. However, as their presentation is still not clear enough to me and some points raised by other reviewers, e.g. computation complexity vs performance improvement are not well explained, I am inclined to keep my current rating.

**Summary Of The Paper:**

This paper proposes a local augmentation strategy to enhance the graph representation and benefits corresponding downstream tasks, where the augmentation is achieved with a generation conditioned on the local neighbor structure. The authors justify the proposed method with both quantitative and qualitative results on benchmark datasets.

**Summary Of The Review:**

This paper presents an interesting local augmentation method in a way of generation to enhance the graph representations and show the results of classification with different backbones on different datasets and we can see the improvement of local augmentations, but can be further improved by providing more ablation studies.

---

> ### Author Response · Authors · 2021-11-16
> **Response to Reviewer cVVh (Part 1)**
>
> We thank the reviewer for recognition of several key elements of our paper:
> 1. The key contribution/novelty: performing feature-level augmentation on graph by learning a distribution conditioned on local structure and the center node's features, via a generative model.
> 2. Clear representation of the motivation and technical part
> 3. Convincing experimental evidence of the our method's superiority compared to exisiting methods!
>
>  **Please refer to our general response about the intuitions and contributions of our paper.**
>
>
>
>
>
> **Q1: However, I still have some small confusion about the notations in section 2.2. For example: Is $p_{\psi}\left(z \mid X_{i}\right)$ a parameterized by $\psi$ or not? In the text I found $p_{\psi}\left(z \mid X_{i}\right)=N(0, I)$.**
>
>
>
> A1: Yes, it is parameterized by $\psi$, we follow the rigorous mathematical derivation in [1]. As you can see the Equation(3) in the paper [1], $p_{{\theta}}(\mathbf{z})$ is also parameterized by $\theta$. Max Welling assumes it is a standard normal distribution. In fact, in VAEs, we usually assume that this item is a standard normal distribution.
>
>
>
>
>
> **Q2: The experiments show the results of classification with different backbones on different datasets and we can see the improvement of local augmentations. Some expected ablations are not shown.**
>
>
>
> A2: We do provide lots of ablation studies in the experiments section, and the ablation studies show that our method is effective. If you mean other ablation studies, please let us know.
>
>
>
>
>
> **Q3: I guess it is non-trivial to re-organize Table 3, as it would be easier to compare if the authors could put methods on the same backbone together (e.g. GCN, G-GCN, DE-GCN, GAUG-OGCN and LA-GCN).**
>
>
>
> A3: We have reorganized this table based on your suggestion. Please see our new submission of the paper.
>
>
>
>
>
> **Q4: If it is possible, it will be better to show the results of GAT, GCNII and GRAND with the other augmentation methods.**
>
>
>
> A4: OK, we give you the results of the GAT, GCNII, and GRAND with other augmentation methods.
>
> | Model | Cora |  Citeseer | Pubmed |
> |:---:|:---:|:---:|:---:|
> |GCN | 81.6 | 70.3  |78.9 |
> |G-GCN |83.7  |71.3 | 80.9 |
> |DropEdge-GCN  |82.8|  72.3| 79.6 |
> |GAUG-O-GCN | 83.6|  73.3|  79.3 |
> |LA-GCN | 84.1 | 72.5|  81.3 |
> |GAT| 83.0 |70.4 |OOM |
> |G-GAT |61.8 | 40.2| OOM |
> |DropEdge-GAT | 82.3  |70.5 |OOM |
> |GAUG-O-GAT |82.2 | 71.6 |OOM|
> |LA-GAT |83.9 |72.3 |OOM |
> |GCNII |85.2 | 73.1 | 80.0 |
> |G-GCNII |35.6 |39.0| 52.6 |
> |DropEdge-GCNII |83.1 | 69.0 | 79.9|
> |LA-GCNII |85.2 | 73.7|  81.6|
> |GRAND |85.4 |75.4 | 82.7 |
> |DropEdge-GRAND|  84.5 |74.4| 80.9 |
> |LA-GRAND| $\mathbf{85.8}$| $\mathbf{75.8}$|  $\mathbf{83.3}$ |
>
>
>
> The results of GAUG-O-GAT and DropEdge-GRAND are taken from corresponding papers. The data augmentation method of G-GNN can't be applied to GRAND, since GRAND uses MLP as architecture and G-GNN must take GNN as architecture. Besides, we don't do experiments on GAUG-O-GCNII and GAUG-O-GRAND since its augmentation method is similar to DropEdge, which deletes some edges to get a sparse graph. As the results show, our augmentation method is significantly stronger than other methods. I think it has two reasons. Firstly, as we know, GNN uses the neighbors in the second-order range to aggregate features. Although GCNII is a 64-layer architecture, it also uses initial residual to construct a skip connection from the input layer to preserve the locality of the node representations. So the locality of node representations is so important for GNNs. Secondly, our method does not rely on any specific architecture and can be applied to any GNN model in a plug-and-play manner. It is these two reasons that ensure that our model achieves great success on different datasets. To our best knowledge, our model achieves new state-of-the-art performance on cora, citeseer, and pubmed.
>
>
>
>
>
> **Q5: I am curious how the proposed local augmentation performs when combined with other augmentation techniques. For example, will it further improve the GNNs by putting LA and DropEdge together?**
>
>
>
> A5: Our method is feature-level augmentation, which may be combined with topology augmentation. But I think this is essentially a sampling problem, that is to say, for the central node, how many neighbors do we need to sample for CVAE training. And we also discuss it in the conclusion section. And one future work is that we can extract more 2/3-hop neighbors if the central node’s degree is small and learn the conditional distribution for random sampling nodes if the graph is large. In addition, we also need to ensure that the neighbors sampled have enough diversity. So the sampling strategy is so important since it can help us to learn the conditional distribution better.

---

> > ### Author Response · Authors · 2021-11-16
> > **Response to Reviewer cVVh (Part 2)**
> >
> > **Q6: How does the importance sampling module improve the generated feature quality? Probably it could be shown in Fig 3 to justify the importance of the proposed module.**
> >
> >
> >
> > A6: We just follow the paper [2] to determine the optimal inference samples to augment the training set. This paper [2] also lacks theoretical analysis. But as this paper explains, importance sampling/active learning can help improve the diversity of inference samples. And Fig 3 only shows that our inference feature matrix follow the distribution of the initial feature matrix.
> >
> >
> >
> >
> >
> > [1] Diederik P. Kingma, Max Welling. Auto-Encoding Variational Bayes. In Advances in Neural Information Processing Systems, 2014.
> >
> > [2] Christopher Nielsen and Michal M Okoniewski. Gan data augmentation through active learning inspired sample acquisition. In CVPR Workshops, pp. 109–112, 2019.

---

> > > ### Author Response · Authors · 2021-11-27
> > > **Do you have any questions?**
> > >
> > > Hi,
> > >
> > >     Your suggestion helps us to improve the quality of the paper, and could you have any concerns?
> > >     We are very happy to answer any questions.
> > >
> > > Best,
> > >
> > > Paper67 Authors.

---

> > > > ### Author Response · Authors · 2021-12-07
> > > > **Response to your comments to our rebuttal**
> > > >
> > > > **After carefully reading all the reviews, the authors' responses, and corresponding revisions, I think the quality of this paper is better than the original submission. However, as their presentation is still not clear enough to me and some points raised by other reviewers, e.g. computation complexity vs performance improvement are not well explained, I am inclined to keep my current rating.**
> > > >
> > > > You first said, **"The presentation of the technical part is generally well written and clear"**. But after reading our response and other reviews, you said, **"However, as their presentation is still not clear enough to me"**. We feel confused about this. We appreciate you very much if you can point out the details of the presentation that you are not clear about.
> > > >
> > > >
> > > > **"computation complexity vs performance improvement"**
> > > >
> > > > As we explained before, the computation complexity is $n\times O(|\mathcal{E}|)$, where $|\mathcal{E}|$ is the number of edges on a graph. For GCN, $n$ is 200, and for our augmentation method, it is about 20.
> > > >
> > > > For the performance improvement, as you can see the results from the table in our rebuttal to your review, our performance is better than other data augmentation methods for all backbones except compared with GAUG-O on GCN in Citeseer. Even combined with GAT, GCNII, and GRAND, the performance of other data augmentations methods would drop. But our method can consistently improve the performance for various backbones. Besides, we can achieve a competitive performance for GCN with our data augmentation method.

---

### Official Review · Reviewer_Dob2 · 2021-11-04

**Correctness:** 3
**Technical Novelty And Significance:** 2
**Empirical Novelty And Significance:** 2
**Recommendation:** 5
**Confidence:** 3

**Main Review:**

Generally, the idea is clear and easy to follow. The proposed local augmentation is feature-level augmentation, which uses CVAE to generate the neighboring features based on the local structure and the center node's features. The augmented features can 1) improve the generalization and 2) recover the noisy/missing features. However, several issues require addressing as listed below.

1. The unclear comparison with baselines. GraphVAE can actually generate the structure and node features by VAE (\tilde{A} and \tilde{F} in the paper). It is suggested to discuss the main difference. Moreover, in Section 2.2, it states that the feature-level augmentation \bar{X} is from other papers [Kong et al., 2020]. It is unclear that this paper modifies [Kong et al., 2020] or directly applies it. Last, the idea of using Skip-gram in graph embedding has been developed before (e.g., Deepwalk). The difference should be highlighted and the papers should be cited.
2. The theoretical contribution is minor. The optimization of VAE is mature, while there are also several papers using importance sampling for estimating marginal likelihood in VAEs.
3. The improvement to GRAND is relatively small in Table 3, e.g., 0.47% on Cora dataset. Moreover, the experiments on the robustness to missing information should contain other baselines to demonstrate the superiority of the proposed approach.

**Summary Of The Paper:**

This paper studies the problem of feature augmentation for training the graph neural networks. Specifically, given the central nodes features and local structure, the proposed approach estimates the distribution of the node features of neighbors. Experimental results show that the proposed approach can further improve the baseline backbones.

**Summary Of The Review:**

The idea is interesting but the difference and theoretical contributions are unclear. It may be the presentation issue but seems to borrow several parts from other papers. Moreover, the improvement is minor (0.47% on Cora dataset).

---

> ### Author Response · Authors · 2021-11-16
> **Response to Reviewer Dob2 (Part 1)**
>
> Thank you very much for your comments and feedback! **Please refer to our general response about the intuitions and contributions of our paper.**
>
>
>
>
>
> **Q1: The unclear comparison with baselines. GraphVAE can actually generate the structure and node features by VAE (\tilde{A} and \tilde{F} in the paper). It is suggested to discuss the main difference. Moreover, in Section 2.2, it states that the feature-level augmentation \bar{X} is from other papers [Kong et al., 2020]. It is unclear that this paper modifies [Kong et al., 2020] or directly applies it. Last, the idea of using Skip-gram in graph embedding has been developed before (e.g., Deepwalk). The difference should be highlighted and the papers should be cited.**
>
>
>
> A1: **"GraphVAE can actually generate the structure and node features by VAE (\tilde{A} and \tilde{F} in the paper). It is suggested to discuss the main difference."**
>
> First of all, our main contribution is not on the Graph VAE. The generator can be replaced with other generative models such as GAN [1] or EBM [2]. We only use VAE to learn the conditional distribution. And our main contribution is local augmentation. The method of our paper is to augment neighbors, which can increase the richness of neighbors' feature representations. As we all know about GNN models, stacking more layers tends to degrade the performance of these models. In other words, we mainly use nodes in the local neighborhood for message passing. But the number of nodes in the local neighborhood is limited. A very intuitive idea is our local augmentation method. That is, given the central node’s feature, our local augmentation approach learns the conditional distribution of its neighbors’ features and generates the neighbors’  feature. GraphVAE mainly focuses on graph generation, and our method only uses VAE for a generation.
>
>
>
> **"Moreover, in Section 2.2, it states that the feature-level augmentation \bar{X} is from other papers [Kong et al., 2020]. It is unclear that this paper modifies [Kong et al., 2020] or directly applies it."**
>
> We only give examples and cite these papers that some prior methods generate \bar{X}. And we have removed these citations based on you suggestion. In fact, we do not use any prior method for augmentation. Our method is very novel.
>
>
>
> **"Last, the idea of using Skip-gram in graph embedding has been developed before (e.g., Deepwalk). The difference should be highlighted and the papers should be cited."**
>
> We only use skip-gram to show some similarities with our method to let readers understand our method better.  And our method is not based on/for graph embedding. As you can see from Figure 2(d), our method is to augment neighbors given the central node's feature representation. Besides, we do not use any NLP model in our paper. And we have cited this paper (Deepwalk) and highlighted the differences in the introduction section based on your suggestion. Please see our new submission of the paper.
>
>
>
>
>
> **Q2: The theoretical contribution is minor. The optimization of VAE is mature, while there are also several papers using importance sampling for estimating marginal likelihood in VAEs.**
>
> As our title "Local Augmentation for Graph Neural Networks" explains, we focus on local augmentation for graph data. We only use VAE for augmentation. Our main contribution is not VAE. Using importance sampling is only for improving the efficiency of inference. It's just a trick. In fact, we do not use importance sampling for estimating marginal likelihood in VAEs.
>
>
>
>
>
> **Q3: The improvement to GRAND is relatively small in Table 3, e.g., 0.47% on Cora dataset. Moreover, the experiments on the robustness to missing information should contain other baselines to demonstrate the superiority of the proposed approach.**
>
> To our best knowledge, GRAND achieve state-of-the-art performance in cora, citeseer, and pubmed. It is very difficult to improve performance on the SOTA model, but we make it. In addition, performance improvement also has a marginal diminishing effect. The better performance the model has the less space for improvement. But we can still improve upon GRAND by a margin of 0.4%, 0.4%, and 0.6% on Cora, Citeseer, and Pubmed respectively, and even achieve new state-of-the-art performance. You can see more results about other data augmentation from the response of reviewer cVVh. Other data augmentation methods even reduce performance on the backbones. This shows the superiority of our method. Our method is mainly to learn node representations. In other words, our method mainly compares the performance of node classification tasks with baselines. The experiments on the robustness to missing information only show that our method can complement missing information [3]. In other words, we can exploit the well-learned distribution to complement the contextual information of the local neighborhood to enhance the locality of the node representations. It does not need any other baselines.

---

> > ### Author Response · Authors · 2021-11-16
> > **Response to Reviewer Dob2 (Part 2)**
> >
> > **Q4: The idea is interesting but the difference and theoretical contributions are unclear. It may be the presentation issue but seems to borrow several parts from other papers. Moreover, the improvement is minor (0.47% on Cora dataset).**
> >
> >
> >
> > A4: We have added some notes to explain our intuitions. And we do not borrow any part from other papers.

---

> > > ### Author Response · Authors · 2021-11-25
> > > **Response to Reviewer Dob2 (Part 3)**
> > >
> > > We want to emphasize our contribution.
> > >
> > > In 2019 and 2020, GNN researchers mostly explored the over smoothing issue of GNN. How to train a deep GNN is the focus of research in the last two years. But when GCNII [1] successfully trained a deep model, they also added a residual connection to the input layer. They also emphasized the importance of the locality of node representations in the paper. In other words, local information is the most important for training GNN. Recently, GNN researchers are exploring how to train a good GNN with local sub-structure information. Zeng et al. (2021) [3] pointed out that the key for GNN is to smooth the local neighborhood into informative representation, no matter how deep it is. And they decouple the depth and scope of GNNs to help capture local graph structure. Besides, Zhang et al. (2021) [4] and a submission to ICLR [5] utilize local subgraph structure to learn good representations. Our method explores how to use local information to learn good node representations from another perspective. Our motivation is that one property of the graph is that the number of nodes in the local neighborhood is far fewer than higher-order neighbors. And this property restricts the expressive power of GNNs due to the limited neighbors in the local structure. A very intuitive idea is to use data augmentation to increase the number of nodes in the local substructure. The generative model is just implementation and is not our main contribution. We sincerely hope that our work can contribute to the GNN community. If you have any concerns, please let us know. We are very happy to answer your any question.
> > >
> > >
> > > [1] Ian J. Goodfellow, Jean Pouget-Abadie, Mehdi Mirza, Bing Xu, David Warde-Farley, Sherjil Ozair, Aaron Courville, Yoshua Bengio. Generative Adversarial Nets. In Advances in Neural Information Processing Systems, 2014.
> > >
> > > [2] Will Grathwohl, Kuan-Chieh Wang,  Jorn-Henrik Jacobsen, David Duvenaud, Kevin Swersky, Mohammad Norouzi. Your Classifier is Secretly an Energy Based Model and You Should Treat it Like One. In International Conference on Learning Representations, 2020.
> > >
> > > [3] Junteng Jia and Austion R Benson. Residual correlation in graph neural network regression. In Proceedings of the 26th ACM SIGKDD International Conference on Knowledge Discovery & Data Mining, pp. 588–598, 2020.
> > >
> > > [4] Hanqing Zeng, Muhan Zhang, Yinglong Xia, Ajitesh Srivastava, Andrey Malevich, Rajgopal Kannan, Viktor Prasanna, Long Jin, and Ren Chen. Decoupling the depth and scope of graph neural networks. In Advances in Neural Information Processing Systems, 2021.
> > >
> > > [5] Muhan Zhang, Pan Li. Nested Graph Neural Networks. In Advances in Neural Information Processing Systems, 2021.
> > >
> > > [6] https://openreview.net/forum?id=uxgg9o7bI_3

---

### Official Review · Reviewer_XsGb · 2021-11-06

**Correctness:** 2
**Technical Novelty And Significance:** 2
**Empirical Novelty And Significance:** 2
**Recommendation:** 5
**Confidence:** 2

**Main Review:**

Strengths:
The idea of using generative model to augment GNN training data is interesting.

Weakness:
1. The writing and organization of this paper need to be polished before submission. The presentation is chaotic. There are also grammatical errors and typos through out the paper. eg. 'data argumentation' in Abstract. The experiment numbers across the seven benchmark datasets are hard to read. It is unclear why they are not presented fully in a single table.
2. The description of the proposed methods are unnecessarily notion heavy making it time consuming to decipher what the main ideas really were. Authors may consider elaborating the intuitions of the main ideas clearly before formulating them with notions.
3. Given the complexity of implementing the proposed augmentation methods, the accuracy gain is marginal comparing to the baselines with / without the complex data augmentations. It is unclear if the proposed methods can be useful for real-world applications.


**Summary Of The Paper:**

This paper presents a data augmentation methods for training graph neural networks in general. They firstly fit a generative model that learns the conditional probability of input node features. The training data are then augmented using the generative model using the proposed importance sampling method and used for training GNNs. The proposed methods were evaluated on 7 public graph benchmark datasets. Authors attempted to show that the proposed augmentation methods could improve the benchmark classification accuracy over the baseline methods.

**Summary Of The Review:**

Though the idea of using generative model to augment GNN training data sounds interesting, I would not recommend to accept this paper given its current presentation quality.

---

> ### Author Response · Authors · 2021-11-16
> **Response to Reviewer XsGb (Part 1)**
>
> Thank you very much for your comments and feedback! **Please refer to our general response about the intuitions and contributions of our paper.**
>
> **Q1: The writing and organization of this paper need to be polished before submission. The presentation is chaotic. There are also grammatical errors and typos through out the paper. eg. 'data argumentation' in Abstract. The experiment numbers across the seven benchmark datasets are hard to read. It is unclear why they are not presented fully in a single table.**
>
> A1: **"The writing and organization of this paper need to be polished before submission. The presentation is chaotic."**
>
> We have reorganized our paper based on your suggestion. We added a new section "Discussion" and placed the part "connection to EP-P and GraphSAGE" in this section. And we only described our method in the method section without any discussion. Besides, we added a new section "Background" to discuss notations and prior graph data augmentation works to let readers know prior work and the difference of our work with prior work. And we placed the related work section before the conclusion section.
>
> **"There are also grammatical errors and typos through out the paper. eg. 'data argumentation' in Abstract."**
>
> We are very sorry that we make a typo error in the abstract section. To be honest, some of our authors come to detect grammatical errors sentence by sentence. But we do ignore the abstract section. If you find more grammatical errors and typos, please let us know. We have modified the corresponding part based on your suggestion. Please see our new submission of the paper.
>
> **"The experiment numbers across the seven benchmark datasets are hard to read. It is unclear why they are not presented fully in a single table."**
>
> We reorganized Table 2, 3. We present results in two tables for two reasons. First, Cora, Citeseer, Pubmed are the three classical datasets of our GNN community for node classification evaluation. We generally follow the dataset split setting by Yang et al. [1]. Results for all the baseline methods are taken from the corresponding papers except GCN, GAT, GCNII, and GRAND since we use these four models as the backbones and need to do experiments for these models. Second, for Squirrel, Actor, Chameleon, and Cornell, there is no standard dataset split. Many papers use their own splits, so we can’t take their results directly from these papers. And Shchur et al. [2] confirm the need for evaluation strategies based on multiple splits. So for a solid evaluation, we do experiments based on 10 random splits for these four datasets. Besides, Cora, Citeseer, and Pubmed are from [1], and Squirrel, Actor, Chameleon, and Cornell are from [3]. The source and the splits of these datasets are different. So we can't present in a single table. And we do experiments on these datasets for two advantages. On the one hand, we inherit the settings of three classical datasets and can compare our method with many previous methods. The results show that we achieve state-of-the-art performance. On the other hand, for solid evaluation, we adopted the suggestion of [3] and do experiments with 10 random dataset splits. The results show that our method can improve performance consistently over all datasets.
>
> **Q2: The description of the proposed methods are unnecessarily notion heavy making it time consuming to decipher what the main ideas really were. Authors may consider elaborating the intuitions of the main ideas clearly before formulating them with notions.**
>
> A2: We are sorry for the trouble with your reading. We have added a high-level view of our method before describing it. And we have added some notes in the abstract section and the part "Complementing missing information" to elaborate the intuitions of our idea based on your suggestion. Please refer to our new submission of our paper. Let's explain the notations equation by equation. Eq. (2) is the general probability notation of all GNN models. We believe that the performance of GNN can be improved through data augmentation, so we have Eq. (3). And the data augmentation is done based on A and X. Eq. (4) is an important technique that decomposes the probability notation of data augmentation and GNN. There are two benefits in the decomposition. First, it allows us to decouple the training of the downstream predictor P and the generator Q, enabling the generator to easily generalize to other downstream tasks. Eq. (5) and Eq. (6) are the derivation of CVAE, here it is emphasized that our method is not a general data augmenatation, we fully consider the neighbor information. It is these two equations that highlight our approach. Eq. (7) is the probability equation for optimizing GNN, which also can be seen in the GCN paper. Eq. (8) and Eq. (9) show the details of the downstream GNN models. Eq. (10) is our trick to optimize the overall training. To conclude, I think it is necessary to use mathematical notations to demonstrate our method.

---

> > ### Author Response · Authors · 2021-11-16
> > **Response to Reviewer XsGb (Part 2)**
> >
> > **Q3: Given the complexity of implementing the proposed augmentation methods, the accuracy gain is marginal comparing to the baselines with / without the complex data augmentations. It is unclear if the proposed methods can be useful for real-world applications.**
> >
> >
> > A3: **"Given the complexity of implementing the proposed augmentation methods, the accuracy gain is marginal comparing to the baselines with / without the complex data augmentations"**
> >
> > Our data augmentation method does not have high computation complexity. The computation complexity of our augmentation method and GCN is related to the size of a graph, such as the number of nodes and edges. One epoch of our augmentation method training is the same as one epoch of GCN training in computation complexity, but we can only use 10 epochs to complete pre-training. The cost of our augmentation method can even be ignored. As you can see from the results, we can improve upon GCN by a margin of 2.5%, 2.2%, and 2.4% on Cora, Citeseer, and Pubmed respectively. As we all know, GCN is a simple model. But combined with our data augmentation method, it can achieve competitive performance. Besides, compared with other data augmentation methods, our method is also superior to them. More importantly, our method can be applied to any GNN model in a plug-and-play manner. So our proposed framework can be equipped with various popular backbone networks such as GCN, GAT, GCNII, and GRAND, which are four GNN models with different characteristics and emphasizes different aspects of GNN models. To our best knowledge, GRAND achieve state-of-the-art performance in cora, citeseer, and pubmed. It is very difficult to improve performance on the SOTA model, but we make it. In addition, performance improvement also has a marginal diminishing effect. The better performance the model has the less space for improvement. But we can still improve upon GRAND by a margin of 0.4%, 0.4%, and 0.6% on Cora, Citeseer, and Pubmed respectively, and even achieve new state-of-the-art performance. You can see more results about other data augmentation from the response of reviewer cVVh. Other data augmentation methods even reduce performance on the backbones. This shows the superiority of our method.
> >
> >
> >
> > **"It is unclear if the proposed methods can be useful for real-world applications."**
> >
> > It is clear that our method can be useful for real-world applications. One potential application is drug discovery. Given a molecule, we can learn the conditional distribution of the groups around the reaction center [4]. And by doing this, we can generate more reasonable molecules for downstream training. It is also a future work for us.

---

> > > ### Author Response · Authors · 2021-11-25
> > > **Response to Reviewer XsGb (Part 3)**
> > >
> > > We think that the focus of your review is the quality of our presentation. We have made a lot of changes in our latest submission according to your suggestion. We think the motivation and intuition of our paper are more important. In 2019 and 2020, GNN researchers mostly explored the over smoothing issue of GNN. How to train a deep GNN is the focus of research in the last two years. But when GCNII [1] successfully trained a deep model, they also added a residual connection to the input layer. They also emphasized the importance of the locality of node representations in the paper. In other words, local information is the most important for training GNN. Recently, GNN researchers are exploring how to train a good GNN with local sub-structure information. Zeng et al. (2021) [5] pointed out that the key for GNN is to smooth the local neighborhood into informative representation, no matter how deep it is. And they decouple the depth and scope of GNNs to help capture local graph structure. Besides, Zhang et al. (2021) [6] and a submission to ICLR [7] utilize local subgraph structure to learn good representations. Our method explores how to use local information to learn good node representations from another perspective. Our motivation is that one property of the graph is that the number of nodes in the local neighborhood is far fewer than higher-order neighbors. And this property restricts the expressive power of GNNs due to the limited neighbors in the local structure. A very intuitive idea is to use data augmentation to increase the number of nodes in the local substructure. The generative model is just implementation and is not our main contribution. We sincerely hope that our work can contribute to the GNN community. If you have any concerns, please let us know. We are very happy to answer your any question.
> > >
> > >
> > > [1] Zhilin Yang, William Cohen, and Ruslan Salakhutdinov. Revisiting semi-supervised learning with graph embeddings. In International Conference on Machine Learning (ICML), 2016.
> > >
> > > [2] Oleksandr Shchur, Maximilian Mumme, Aleksandar Bojchevski, Stephan Günnemann. Pitfalls of Graph Neural Network Evaluation. Workshop on Relational Representation Learning. NeurIPS, 2019.
> > >
> > > [3] Hongbin Pei, Bingzhe Wei, Kevin Chen-Chuan Chang, Yu Lei, Bo Yang. Geom-GCN: Geometric Graph Convolutional Networks. In International Conference on Learning Representations, 2020.
> > >
> > > [4] Chence Shi, Minkai Xu, Hongyu Guo, Ming Zhang, Jian Tang. A Graph to Graphs Framework for Retrosynthesis Prediction. In International Conference on Machine Learning, 2020.
> > >
> > > [5] Hanqing Zeng, Muhan Zhang, Yinglong Xia, Ajitesh Srivastava, Andrey Malevich, Rajgopal Kannan, Viktor Prasanna, Long Jin, and Ren Chen. Decoupling the depth and scope of graph neural networks. In Advances in Neural Information Processing Systems, 2021.
> > >
> > > [6] Muhan Zhang, Pan Li. Nested Graph Neural Networks. In Advances in Neural Information Processing Systems, 2021.
> > >
> > > [7] https://openreview.net/forum?id=uxgg9o7bI_3

---

### Author Response · Authors · 2021-11-12
**Changes since the original submission 11/12/2021**

We thank the reviewers for their thorough reviews and useful comments! We feel that they have helped us improve the paper.

We updated our submission and included the following main changes:

1.  Fixed a typo error in the abstract section.
2. Highlighted the difference between our paper and Deepwalk.
3. Organized table 3.
4. Renamed section 2.3.
5. Added some clarification about Eq.(6).

---

### Author Response · Authors · 2021-11-14
**Changes since the original submission 11/14/2021**

We updated our submission and included the following main changes:

1. Added some notes to explain the intuition of our paper in the introduction section.
2. Corrected errors in Eq.(4) and Eq.(8)
3. Reorganized algorithm 1.
4. Reorganized section 2.4

---

### Author Response · Authors · 2021-11-16
**Changes since the original submission 11/16/2021**

We updated our submission and included the following main changes:

1. Added some notes to explain the intuition of our paper in the introduction section.
2. Added some notes to explain the intuition of the paper "Complementing missing inform".
3. Added some notes to show our method is different from graph/word embeddings.
4. Reorganized the paper.
    1). Added a new section background. Placed the topology-level and feature-level augmentation of the related work section in 11/14 submission to the background section.
    2). Placed the related work section before the conclusion section
    3). Added a new section discussion. Placed "connection to EP-B and GraphSAGE" to the discussion section.

---

### Author Response · Authors · 2021-11-16
**General response to the reviews**

Thank you very much for your comments and feedback!

The intuitions and contributions of our paper are following.

Before answering the questions, we need to list some papers to illustrate our contribution. Our main contribution is local augmentation, as the title shows.

Over-smoothing is an important problem in our GNN community, firstly introduced by Li et al. [1]. It means stacking more layers tends to degrade the performance of GNN models. Why do we use high-layer GNN models, the performance will become bad? Because all node feature representations will become indistinguishable after multiple layers of graph convolution, as explained by Chen et al. [4]. We think our paper solves a very fundamental problem for GNNs. Prior works, such as GCNII and JKnet, emphasize the importance of the locality of node representations. Recent work[11] pointed out that the key for GNN is to smooth the local neighborhood into informative representation, no matter how deep it is. And we can see that the local information is so significant to learn node representations. But we all know that the number of nodes in the local neighborhood is far fewer than higher-order neighbors, which restricts the expressive power of GNNs due to the limited neighbors in the local structure. We propose a very novel technique to augment nodes in the local neighborhood, which provides enough local information for various GNNs to learn good node representations.

Our main contribution is to do local augmentation based on the generative model. The generative model is not our main contribution. Even CVAE can be replace by GAN[7] or EBM[8]. CVAE only learns s the conditional distribution of the connected neighbors’ representations given the representation of the central node. Extensive experiments and analyses show that local augmentation consistently yields performance improvement for various GNN architectures across a diverse set of benchmarks. This also shows that our intuition is correct, and it makes a great contribution to the entire GNN community. In addition, compared with two famous works EP-B [9] and GraphSAGE [10], our method is the third paradigm to exploit neighbors. This is an important breakthrough. Then we answer your questions.

[1] Qimai Li, Zhichao Han, and Xiao-Ming Wu. Deeper insights into graph convolutional networks for semi-supervised learning. In Proceedings of the AAAI Conference on Artificial Intelligence, 2018.

[2] Thomas N. Kipf and Max Welling. Semi-supervised classification with graph convolutional networks. In International Conference on Learning Representation, 2017.

[3] Yu Rong, Wenbing Huang, Tingyang Xu, and Junzhou Huang. Dropedge: Towards deep graph convolutional networks on node classification. In International Conference on Learning Representation, 2020.

[4] Ming Chen, Zhewei Wei, Zengfeng Huang, Bolin Ding, and Yaliang Li. Simple and deep graph convolutional networks. In International Conference on Machine Learning, pp. 1725–1735. PMLR, 2020.

[5] Keyulu Xu, Chengtao Li, Yonglong Tian, Tomohiro Sonobe, Ken-ichi Kawarabayashi, and Stefanie Jegelka. Representation learning on graphs with jumping knowledge networks. In International Conference on Machine Learning, pp. 5453–5462. PMLR, 2018.

[6] Petar Veličković, Guillem Cucurull, Arantxa Casanova, Adriana Romero, Pietro Liò, Yoshua Bengio. Graph attention networks. In International Conference on Learning Representations, 2018.

[7] Ian J. Goodfellow, Jean Pouget-Abadie, Mehdi Mirza, Bing Xu, David Warde-Farley, Sherjil Ozair, Aaron Courville, Yoshua Bengio. Generative Adversarial Nets. In Advances in Neural Information Processing Systems, 2014.

[8] Will Grathwohl, Kuan-Chieh Wang,  Jorn-Henrik Jacobsen, David Duvenaud, Kevin Swersky, Mohammad Norouzi. Your Classifier is Secretly an Energy Based Model and You Should Treat it Like One. In International Conference on Learning Representations, 2020.

[9] Alberto García-Durán and Mathias Niepert. Learning graph representations with embedding propagation. In Advances in Neural Information Processing Systems, 2017.

[10] William L Hamilton, Rex Ying, and Jure Leskovec. Inductive representation learning on large graphs. In Advances in Neural Information Processing Systems, 2017.

[11] Hanqing Zeng, Muhan Zhang, Yinglong Xia, Ajitesh Srivastava, Andrey Malevich, Rajgopal Kannan, Viktor Prasanna, Long Jin, Ren Chen. Decoupling the Depth and Scope of Graph Neural Networks. In Advances in Neural Information Processing Systems, 2021.

---

### Author Response · Authors · 2021-11-19
**Changes since the original submission 11/19/2021**

We updated our submission and included the following main changes:

1. Added a high-level view of our method in "Local Augmentation" section.
2. Reorganized the table 3.
3. Deleted some sentences in the related work section.
4. Some minor modifications in the abstract section and the part "Connection to EP-B and GraphSAGE".

---

### Author Response · Authors · 2021-11-22
**Changes since the original submission 11/21/2021**

We updated our submission and included the following main changes:

1. Rewrote the abstract, introduction, and conclusion sections.
2. We emphasized the importance of local augmentation from different perspectives.

---

### Decision · Program_Chairs · 2022-01-20

**Decision:**

Reject

**Comment:**

The paper presents a new algorithm for data augmentation in graph neural networks. The algorithm works by learning a conditional model of a node's neighbor features, and augment the neighborhood representation using the generative model.

 In response to the reviews, the authors provided long answers and clarified much of the text. Nonetheless, after the discussion, two main concerns remained. First, the presention still felt subpar, too notationally heavy for what was presented. Second, the gains with respect to the baselines were assessed as not sufficiently significant to justify the approach which is substantially more complex than a baseline such as GRAND.